# What Mechanisms Does Knowledge Distillation Distill?

**Cindy Wu**[+]
University of Cambridge

**Ekdeep Singh Lubana**
University of Michigan
CBS, Harvard University

**Bruno Kacper Mlodozeniec**
University of Cambridge

**Robert Kirk**
University College London

**David Krueger**
University of Cambridge

## Abstract

Knowledge distillation is a commonly-used compression method in ML due to the popularity of increasingly large-scale models, but it is unclear if all the information a teacher model contains is distilled into the smaller student model. We aim to formalize the concept of 'knowledge' to investigate how knowledge is transferred during distillation, focusing on shared invariant outputs to counterfactual changes of dataset latent variables (we call these latents mechanisms). We define a student model to be a good stand-in model for a teacher if it shares the teacher's learned mechanisms, and find that Jacobian matching and contrastive representation learning are viable methods by which to train such models. While these methods do not result in perfect transfer of mechanisms, we show they often improve student fidelity or mitigate simplicity bias (as measured by the teacher-to-student KL divergence and accuracy on various out-of-distribution test datasets), especially on datasets with spurious statistical correlations.

## 1 Introduction

Increasingly large deep neural networks (DNNs) trained on huge, web-crawled datasets have shown unprecedented performance on a multitude of tasks [1, 2, 3, 4, 5, 6, 7, 8, 9, 10, 11, 12, 13, 14, 15, 16, 17, 18, 19, 20, 21, 22, 23, 24, 25], including emergent capabilities that help with more general-purpose and flexible behaviour with SOTA on tasks they were not finetuned on [26] [11] [18]. However, resource constraints faced in realistic scenarios, e.g., latency or energy budgets, impact the feasibility of practically deploying such large models. *Knowledge distillation* [27, 28, 29, 30, 31, 32, 33, 34] was motivated as a framework to address this challenge, wherein a smaller "student" model is trained to mimic the outputs produced by the pre-trained "teacher" network on some available dataset. The underlying hypothesis is that enforcing consistency between the outputs produced by two models will yield a "transfer of knowledge" [27], resulting in the less performant model (student) inheriting the *mechanisms* [35] used by the more performant model (teacher) to make its predictions.

The immense success of distillation in several diverse domains [28, 29, 30, 31, 36, 37, 38] does make the argument above sound intuitively correct. However, follow-up work focused on developing a better understanding of knowledge distillation has raised doubt on this viewpoint [39, 40, 41, 42, 31, 43, 44, 45]. These works demonstrate that distilled student models infrequently make the same errors as the teacher models. This is an unlikely result if the models were sharing knowledge (and hence relying on the same mechanisms for making their predictions). These papers make the success of knowledge distillation surprising, and it remains unclear what precise prediction mechanisms, if any, the student inherits from the teacher. Since the student is often observed to outperform the teacher [27, 39, 46, 45, 47], it may be learning entirely novel mechanisms that the teacher does not

---

[+]Correspondence to `wu.cindyx@gmail.com`.

37th Conference on Neural Information Processing Systems (NeurIPS 2023).

even possess. This is possible because in practical settings, the distillation dataset is likely of direct relevance to the application of interest. It is also likely smaller than (or minimally overlaps with) the teacher's pretraining data, and may not be available in offline distillation [27, 48, 49, 50, 51]. Since these smaller distillation datasets are often underspecified (i.e. they contain several predictive attributes that can be used to produce the correct output [52, 53, 54, 55, 56]), a student can in principle learn to match outputs produced by the teacher through a different mechanism to that used by the teacher. Previous explorations into why knowledge distillation works suggests it is unclear what would motivate the student to learn similar prediction mechanisms as the teacher. For example, Cheng et al. [41] suggest that knowledge distillation enforces learning various concepts simultaneously. In addition to providing additional information, they find that teacher outputs guide the optimisation process by preventing excessive exploration of the loss landscape. Phuong et al. [40] find data geometry (e.g. class separation) and optimiser bias to also be contributing factors. This leads to another question: what design decisions in distillation pipelines incentivize the student to learn the same prediction mechanisms as the teacher model? Beyond developing a better understanding of distillation, answering these questions clarify when distillation can be used for producing a student model that serves as a faithful replacement of its teacher counterpart. We make the following contributions:

- **Formalizing knowledge transfer.** We define successful knowledge transfer as when the student and teacher produce the same outputs under systematically generated counterfactuals of a dataset. This definition abstracts away the precise implementation of a prediction mechanism and only emphasizes the behavioral equivalence of two models to define a notion of 'shared knowledge'.
- **Characterizing knowledge transfer in distillation techniques** Motivated by our definition, we develop synthetic datasets spanning different modalities to allow counterfactual generation and hence enable precise characterization of which prediction mechanisms a model relies on for producing its outputs. We demonstrate that the standard distillation pipeline of matching teacher logits suffers from a *simplicity bias* [57, 58, 59, 60], resulting in the student learning primarily the simplest mechanisms in the distillation dataset. If the distillation dataset and the teacher's pretraining dataset have different distributions, the student and teacher may learn entirely distinct prediction mechanisms.
- **Methods for reducing simplicity bias and improving student-teacher matching.** We investigate two distillation methods aiming to more closely match model representations. We find evidence for decreased teacher-to-student KL divergence and less simplicity bias towards certain *spurious features*.

## 2 Preliminaries: Knowledge Distillation

**Notation.** Consider a neural network $f : \mathbb{R}^n \times \mathbb{R}^d \to \mathbb{R}^K$ that takes $n$-dimensional inputs $x \in \mathcal{X} \subset \mathbb{R}^n$, has parameters $\theta \in \mathbb{R}^d$, and produces an output $f(x; \theta) \in \mathbb{R}^K$ (interpreted as the logits in a classification setting). The neural network predictions are the composition $f^{\text{SM}_T} = \text{softmax}_T \circ f$, where $\text{softmax}_T(z)_i = \frac{\exp z_i/T}{\sum_j \exp z_j/T}$ is the temperature-weighted softmax function for some temperature parameter $T > 0$. Cross-entropy loss on a dataset $\mathcal{D} \in \mathcal{X} \times [K]$ (where $[K]$ denotes the set $\{1, 2, \ldots, K\}$) for a model with parameters $\theta$ is written $\mathcal{L}(f(\mathcal{D}; \theta))$.

Let $\mathcal{D}_{\text{t}}$ be the dataset used to train the teacher model $f_{\text{t}} : \mathbb{R}^n \times \mathbb{R}^{d_{\text{t}}} \to \mathbb{R}^k$ (whose parameters are $\theta_{\text{t}} \in \mathbb{R}^{d_{\text{t}}}$). The goal in knowledge distillation is to use this teacher model to train a 'student' model $f_{\text{s}} : \mathbb{R}^n \times \mathbb{R}^{d_{\text{s}}} \to \mathbb{R}^k$ by finding a set of parameters $\theta_{\text{s}} \in \mathbb{R}^{d_{\text{s}}}$ such that the outputs of the student model $f_{\text{s}}(\cdot; \theta_{\text{s}})$ match, in some specific sense, outputs from the teacher model $f_{\text{t}}(\cdot; \theta_{\text{t}})$ on a 'distillation dataset' $\mathcal{D}_{\text{distill}}$. We distinguish between the dataset used for training the teacher versus the one used for distilling the teacher into the student to a) emphasize the fact that a practitioner who acquires an off-the-shelf, pretrained teacher model (offline training) is unlikely to have access to the data used for training it, and b) explore situations where the student dataset is markedly different - perhaps more diverse and likely containing *spurious mechanisms* [35]. There exists a diverse range of possible distillation methods [28]. Of these, we explore three: Jacobian matching, contrastive representation distillation, and soft targets only from standard distillation. Beyond their widespread use, we choose these methods because they focus only on input/output information – i.e. no intermediate representations are used.

**(A) Standard distillation.** First proposed in the context of neural networks by Hinton et al. [27], the standard distillation pipeline involves optimizing the agreement between teacher and student model predictions by minimizing the $\mathcal{KL}$-divergence between them:

$$\mathbb{E}_{x \sim \mathcal{D}_{\texttt{distill}}} \left[ D_{\mathrm{KL}} \left( f_{\mathtt{t}}^{\mathrm{SM_T}}(x; \theta_{\mathtt{t}}) \| f_{\mathtt{s}}^{\mathrm{SM_T}}(x; \theta_{\mathtt{s}}) \right) \right] \tag{1}$$

The objective above is equivalent to minimising cross-entropy loss with $f_{\mathtt{t}}^{\mathrm{SM_T}}(x; \theta_{\mathtt{t}})$ as the 'soft' targets for a student model. As shown by Hinton et al. [27], assuming the logits $f_{\mathtt{t}}(x), f_{\mathtt{s}}(x)$ are zero-centered (mean zero), in the high temperature limit $T \to \infty$ this is equivalent to minimising the average logit squared difference: $\|f_{\mathtt{t}}(x) - f_{\mathtt{s}}(x)\|^2$.

While a standard cross-entropy loss promoting correct classification is often added to the distillation objective, we follow recent works on understanding knowledge distillation [39, 31] and focus on the distillation objective only. No auxiliary classification loss is added while training the student, isolating the distillation objective's effect in inducing knowledge transfer between teacher and student.

**(B) Jacobian Matching.** A Jacobian matching distillation loss matches norm of the gradient of logits with respect to the input between teacher and student: $f_{\mathtt{t}}^{\mathrm{SM_T}}, f_{\mathtt{s}}^{\mathrm{SM_T}}$ and the input-output Jacobians $\mathbf{J}_{f_{\mathtt{s}}^{\mathrm{SM_T}}(\cdot, \theta_{\mathtt{s}})}, \mathbf{J}_{f_{\mathtt{t}}^{\mathrm{SM_T}}(\cdot, \theta_{\mathtt{t}})} \in \mathbb{R}^{K \times n}$ match on examples in the distillation dataset $\mathcal{D}_{\texttt{distill}}$. This method is equivalent to classical distillation with analytical addition of perturbation noise to inputs. We can decompose the loss function into one representing usual squared error loss and a regularisation term (Tikhonov regulariser). This does not just match on datapoints, but infinitely many points in their neighbourhood. This can be achieved by adding a the following penalty to the standard distillation loss of Eq. 1:

$$\left\| \mathbf{J}_{f_{\mathtt{s}}^{\mathrm{SM_T}}(\cdot, \theta_{\mathtt{s}})}(\boldsymbol{x}) - \mathbf{J}_{f_{\mathtt{t}}^{\mathrm{SM_T}}(\cdot, \theta_{\mathtt{t}})}(\boldsymbol{x}) \right\|_2^2 \tag{2}$$

Beyond distillation [61, 62], the objective above or variants of it have been introduced in several contexts, such as improving out-of-distribution generalization [63], improving adversarial robustness [64], and for learning disentangled representations [65, 66].

**(C) Contrastive Distillation.** In contrastive distillation, the goal is to train the student to maximise the *mutual information* between the representations (typically, the features in the penultimate layer) of the teacher network and the student network on the transfer dataset $\mathcal{D}_{\texttt{distill}}$. In [67], the authors propose to do this by maximising a lower-bound on the mutual information objective given below. Denote by $g_{\mathtt{s}} : \mathbb{R}^n \to \mathbb{R}^{h_{\mathtt{s}}}, g_{\mathtt{t}} : \mathbb{R}^n \to \mathbb{R}^{h_{\mathtt{t}}}$ the functions producing the penultimate layer features in the student and teacher models respectively.

$$
\begin{aligned}
\textbf{Teacher Representation:} \quad & Z_{\mathtt{t}} = g_{\mathtt{t}}(X) \\
\textbf{Student Representation:} \quad & Z_{\mathtt{s}} = g_{\mathtt{s}}(X)
\end{aligned}
\qquad\qquad X \sim \mathcal{D}_{\texttt{distill}}
$$

$$\mathcal{MI}(Z_t, Z_s) \geq \mathbb{E}_{p(Z_{\mathtt{t}}, Z_{\mathtt{s}})}[\log h(Z_{\mathtt{t}}, Z_{\mathtt{s}})] + N \mathbb{E}_{p(Z_{\mathtt{t}})p(Z_{\mathtt{s}})}[\log(1 - h(Z_{\mathtt{t}}, Z_{\mathtt{s}}))] \tag{3}$$

where $h : \mathbb{R}^{h_{\mathtt{t}}} \times R^{h_{\mathtt{s}}} \to [0, 1]$ is a learnable function optimized jointly with the parameters of the student; it can be interpreted as an auxiliary "critic" predicting whether the representations were sampled jointly (from $p(Z_{\mathtt{t}}, Z_{\mathtt{s}})$) or independently (from $p(Z_{\mathtt{t}})p(Z_{\mathtt{s}})$), assuming that they are sampled jointly $1/(N+1)$ of the time. $h$ typically takes the parametric form:

$$h(\boldsymbol{z}_{\mathtt{t}}, \boldsymbol{z}_{\mathtt{s}}) = \frac{\exp \boldsymbol{r}_{\mathtt{t}}^{\mathsf{T}} \boldsymbol{r}_{\mathtt{s}}/\tau}{\exp \boldsymbol{r}_{\mathtt{t}}^{\mathsf{T}} \boldsymbol{r}_{\mathtt{s}}/\tau + \frac{N}{|\mathcal{D}_{\texttt{distill}}|}}, \qquad\qquad \begin{aligned} \boldsymbol{r}_{\mathtt{t}} &= W_{\mathtt{t}} \boldsymbol{z}_{\mathtt{t}}/\|W_{\mathtt{t}} \boldsymbol{z}_{\mathtt{t}}\|, \\ \boldsymbol{r}_{\mathtt{s}} &= W_{\mathtt{s}} \boldsymbol{z}_{\mathtt{s}}/\|W_{\mathtt{s}} \boldsymbol{z}_{\mathtt{s}}\|, \end{aligned} \tag{4}$$

where $W_{\mathtt{s}} \in \mathbb{R}^{h_{\texttt{inter}} \times h_{\mathtt{s}}}, W_{\mathtt{t}} \in \mathbb{R}^{h_{\texttt{inter}} \times h_{\mathtt{t}}}$ are learnable parameters, and $\tau, h_{\texttt{inter}}$ are hyperparameters.

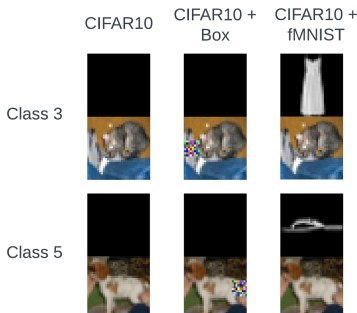

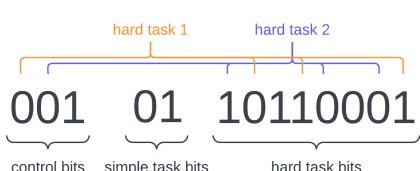

Figure 1: **Synthetic Datasets.** Following the protocol above, we embed synthetic cues in two existing datasets: (1) Dominoes [57], where CIFAR-10 images are concatenated with F-MNIST images with the same class number. A third spurious mechanism selects a location of the CIFAR-10 image to set as randomised pixels (we refer to this as 'box'). If the box mechanism is correlated with CIFAR-10, then the location is determined by the CIFAR-10 label. (2) Spurious parity, where the simple task acts as a spurious mechanism. The label is the parity of the specified hard task subsequence (which is selected for by the control bit). All the hard tasks together act as a single complex mechanism. Shown here is a setup with 3 hard tasks, 8 total hard task bits, 3 hard task bits per task and 2 simple task bits.

## 3 Defining Knowledge Transfer

**Motivation.** As discussed in Sec. 1, despite distillation's immense success in various fields [28, 29, 36, 30, 31, 37], a formal notion of precisely what knowledge, if any, is transferred from the teacher to student has yet to be defined. For instance, consider a visual object recognition task. In such scenarios, backgrounds are often correlated with the object category due to sampling bias [52, 53, 56]. Here, a model can rely on either the background or more intrinsically meaningful attributes of the object, such as its shape, to solve the recognition task. To understand which, we can evaluate how the model's prediction changes when image backgrounds are altered. If predictions change, the model relies on information in the (spurious) attribute of image background; if the predictions do not change, the model is invariant to background. Formalizing this intuition, prior work calls use of a predictive attribute to produce outputs a "mechanism" [35], and defines two models that rely on the same mechanisms as *mechanistically similar*. This framework is relevant to the

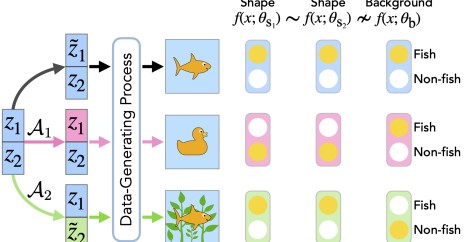

Figure 2: **Knowledge Transfer:** We define successful knowledge transfer of a teacher and student model based on how they respond to unit interventions on the data-generating process, i.e., interventions on specific dimensions of the latent vector $z$; e.g., $\mathcal{A}_1$ (shape) and $\mathcal{A}_2$ (background) in the figure. Here, yellow circles represent the prediction of a given model (column) on a counterfactual image (row). Models whose predictions are invariant to the same set of interventions (denoted $\theta_1 \sim \theta_2$) are termed mechanistically similar.

problem of knowledge transfer in distillation as well. Specifically, if a student model that is perhaps more resource-efficient behaves the same way as a teacher model (i.e. is mechanistically similar), it can serve as a *faithful* replacement of the teacher. We generalize their formalization to our distillation setup next.

Let $I = (i_1, \ldots, i_k)$ denote a non-empty subsequence of indices $(1, 2, \ldots, d)$. Consider a set of latents $z \in \mathcal{Z}$, that instantiates a data-generating process (DGP) $g : \mathcal{Z} \to \mathcal{X}$ from the latents $z$ to observations $x$ and a labeling function $h : \mathcal{Z} \to \mathcal{Y}$ from latents $z$ to labels $y$. We assume *observational sufficiency* of the DGP: observations are sufficient for determining the label.

**Definition 3.1. (Mechanism.)** For a particular latent configuration $z \in \mathcal{Z}$, we say that $f(.; \theta) : \mathcal{X} \to \mathcal{Y}$ uses mechanism $I$ on that example (where $I \subseteq [d_z]$ is the subset of indices of the latents) whenever $f(g(z'); \theta) = f(g(z); \theta)$ for all $z' \in \mathcal{Z}|z'_I = z_I$.

Based on previous research (Appendix A), simplicity bias is where a model tends to rely on latent features which produce a 'simpler' decision boundary or solution. We call such latents corresponding to spurious correlations in the learned model *spurious cues*.

While a few recent analyses have addressed similar questions, they focus on in-distribution test datasets and coarse-grained measures like loss values [39, 31, 44]. In contrast, we emphasize out-of-distribution, counterfactual datasets. This is motivated from a notion of behavioral equivalence as the ideal goal of knowledge distillation. Our experiments are focused on the following questions for analysis:

1. For soft-target-matching-only distillation, only the simplest mechanisms will match.
2. Training a student model by minimizing the standard distillation objective is insufficient to guarantee knowledge transfer from teacher to student. Meanwhile, there exist distillation methods more likely to transfer all mechanisms.
3. Even if a teacher and student have similar fidelity (accuracy on the base task on the distillation set or even the teacher's training set), they do not necessarily behave the same out of distribution [39].

The second point above is a consequence of the recent advances made in the in the field of Nonlinear Independent Component Analysis [68, 69, 70, 71, 72, 66] and disentangled representation learning [73, 74]. These demonstrate that producing the same outputs on a given dataset is insufficient to guarantee two models rely on the same underlying mechanisms for making their predictions. These suggest distillation is limited in what knowledge it can transfer—this depends on what data is shown to the models during distillation. We probe this further via empirical investigations.

## 4    Mechanistic Evaluation of Distillation

In this section we highlight the experimental setup and high-level results.

### 4.1    Training and Evaluation

**Dataset Generation.** We follow prior work on understanding distillation, which primarily uses synthetic datasets to evaluate distillation protocols in a controlled manner [39, 41, 40, 45, 43, 42]. Having control over the data-generating process allows us to be precise about the distribution shift that occurs in the distillation dataset with respect to the teacher's pretraining data, in order to evaluate a student model's reliance on different mechanisms by altering the underlying latents. We assume a set of 'natural' latents underlie the labeling function $h$ of the data-generating process. All other latents are either uncorrelated with the label or model 'spurious' cues in the data. If using information from spurious latents leads to simpler functions, neural network simplicity bias [59, 57, 58, 35, 75, 76, 77, 78] suggests that a network will rely on them rather than the natural attributes for reducing the task loss. We denote the $n$ mechanisms defined by these spurious latents as $\{I_{s_1}, I_{s_2}, \ldots, I_{s_n}\}$. We design two datasets across both image and text data, called *dominoes* (images) and *parity* (language), shown in Fig. 1. These datasets have been used by prior works for modeling neural networks' behavior regarding simplicity bias [57, 35], transfer learning [79, 80, 81, 82], disentangled representation learning [83], and scaling laws [84].

**Training Protocol.** We use teacher and distillation datasets with different distributions over the latents, modeling the fact that a practitioner training a student model is unlikely to have access to the same dataset as the teacher model's training data. To understand the effect of distribution shifts, we test our dominoes dataset (with an image mechanism and two spurious mechanisms) under all possible combinations of distillation and teacher dataset mechanisms (Figure 3).

**Evaluation Protocol.** To evaluate whether a model uses any given mechanism, we randomise or remove latents corresponding to the content of the original image on the dominoes dataset, and report the expected divergence (as in Definition 3.1). For the image dominoes datasets, we remove the latents entirely.

### 4.2    Distillation Loss and Distribution Shift on Dominoes Dataset

This section explores the effect of distribution shifts on a 3-mechanism image dataset for ResNet-18 self-distillation. We use each of 7 possible datasets where at least 1 of 3 mechanisms exists for teacher training, distillation and test evaluation. This gives 49 student/teacher mechanism combinations and 343 categorical final test values per loss function, each run with 3 seeds. Notation: $S$ is distillation dataset (student) mechanism and $T$ is teacher dataset mechanism. 'Base distillation' means softmax logit KL matching. All mechanisms are denoted by a single letter (see Figure 1) – $I$: image (CIFAR-

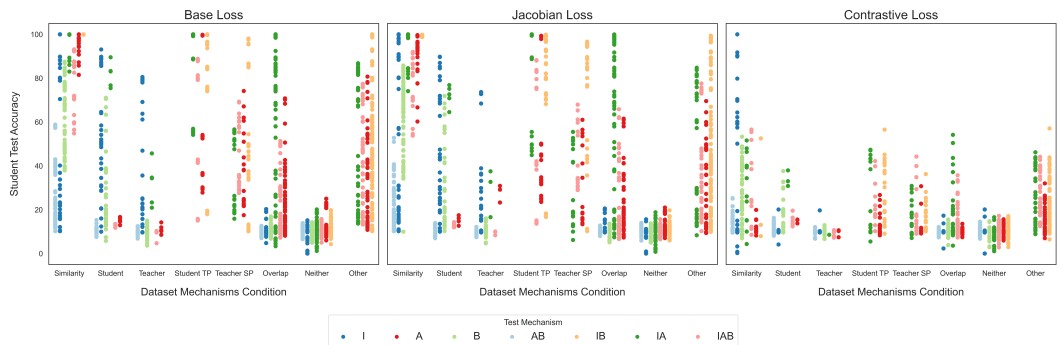

Figure 3: **Final test accuracy. Notation**: *Similarity*: test mechanism overlaps completely with both student and teacher mechanisms. *Student*: test mechanism in $S$ and not in $T$. There must be a shared mechanism between $S$ and $T$, excluding the test mechanism. *Teacher*: test mechanism in $T$ but not in $S$, with same criteria for shared subtask as in Student group. *Student TP*: test mechanism is covered by $S$ and shares a subset with the $T$. *Overlap*: $S, T$ share a subset and this subset is not in test mechanism. *Neither*: teacher shares no mechanisms with student. Any scenarios not fitting these categories are classified under *Other*, a broad class where the student and teacher each contain some subset of the test mechanism. Left: for the 'Similarity' group, a lower bound is observed on performance across all combinations. This bound is highest when all three mechanisms are present (test mechanism IAB). Middle: Jacobian has slightly lower performance when teacher and student do not share mechanisms ('Neither' group), improved performance for certain test mechanisms when only the teacher contains it ('Teacher', 'Teacher SP' groups) and reduced performance for certain test mechanisms when only the student contains it ('Student', 'Student TP' groups). Right: contrastive loss strongly upper bounds test accuracy for spurious mechanisms in 'Similarity' group. Even when the student, teacher and test datasets share the spurious mechanism ('Similarity' group), learning is impeded. The only exception to this is the box mechanism in the test dataset, where simplicity bias is still observed.

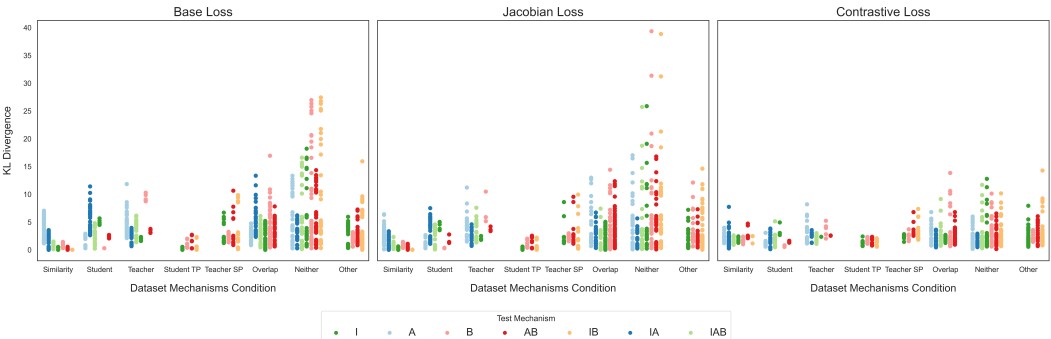

Figure 4: **Final test KL divergence**. Notation is as in Figure 3. Middle: Jacobian loss leads to especially high range of final KL divergence when the teacher and student do not share mechanisms. Right: contrastive loss further bounds teacher-student KL divergence and results in most effective matching of teacher and student.

10), *A*: spurious mechanism A (box), *B*: spurious mechanism B (F-MNIST). In Figures 3, 4, each strip corresponds to a different test mechanism, and each group to the relationship between $S$ and $T$. This grouping does not use the relation between the test mechanism and $S, T$. However, they are used in Appendix D Tables 1, 2. Finally, Figure 9 shows the mean and variance of final accuracy values for separated test mechanisms. Refer to Appendix B, D, E for the rest of the results.

Simplicity bias is observed in Figure 9 column 1 with base distillation. When the box mechanism (mechanism A) is present in the student and teacher datasets, it is learned while CIFAR-10 and F-MNIST are ignored. This is expected behaviour, as the teacher also shows simplicity bias (Section C). Interestingly, the student can learn a new F-MNIST mechanism (column 1, row 4 under test mechanism B) if it is present in the distillation dataset and correlated with the box mechanism—this is an example of a type of 'secondary transfer' which we discuss in further detail below.

For Jacobian loss, data points in Figure 9 with a change greater than 2 standard deviations are often cases of distribution shift. In particular, there is decreased learning if the test mechanism was only in the distillation dataset. Jacobian loss is more likely to match $S$ to $T$ than base distillation if the

test mechanism is in $T$ but not in/partially in $S$ (Table 1 column 'in $T$'). Also, it is less likely to transfer all of $S$ (Table 1 column 'In $S$') if it is not in/partially in the teacher dataset. KL divergence on test datasets decreases if the teacher and student datasets are identical (Figure 13. In contrast, KL divergence increases when $S \neq T$ (e.g. mechanism I and AB) or one of $T$ or $S$ contains extra mechanisms that the other dataset did not (Appendix E). In this sense, Jacobian loss may not improve accuracy, but leads to better matching overall.

Contrastive loss shows strong suppression of box mechanism transfer where it is not present in both teacher and student datasets (Figure 3, Figure 9). This transfer suppression is greatest when the spurious test mechanism is present in only one of the student or teacher datasets. Relative to performance in base distillation, this effect is surprisingly strong for the image corrupted by the box mechanism (test mechanism IA, 'Student'/'Teacher' groups, Figure 3). This may model well behaviour on a type of spurious feature often present in realistic vision datasets. However, for all mechanisms, training with contrastive loss takes longer to achieve the same accuracy, resulting in a significant performance-robustness trade-off (Appendix D, Table 1, column '$EQ$'). It can also lead to transfer of simple mechanisms if they are present in both datasets. Contrastive loss produces the largest decrease in teacher-student output KL divergence compared to base and Jacobian distillation (Figure 4). The greatest KL divergence values for this loss function are for the 'Neither' group, where the teacher and student match but do not match the test mechanism. Contrastive loss seems to trade off matching on the support set of the teacher and student's intersection for poorer performance entirely out of distribution of both.

Base distillation often leads to a type of 'secondary transfer': if the teacher mechanism contains mechanisms P and Q, the student mechanism contains mechanisms Q and R, and we test on just mechanism P, the student may have high performance. This seems obvious, as the shared mechanism Q allows the teacher to produce correctly labelled examples. However, Jacobian and contrastive losses are less likely to produce this effect (Figure 9).

### 4.3 Fraction of Spurious Mechanism on Parity Dataset

Figure 6 shows the training steps required to achieve a given accuracy threshold as a function of the distillation dataset probability of simple task parity correlating with hard task parity. This shows a test dataset where only the hard task corresponds to the label. The case where both simple and hard tasks correspond to the label is in Appendix F. The teacher dataset contains only clean data—i.e. hard task bit parity is the label, while simple task bits are randomised with respect to the label. More hard task substrings to pick from increases the relative difficulty of learning the hard task mechanism, compared to the simple task. For full results, including accuracy and entropy over training time on datasets with counterfactually randomised latents, see Appendix F.

Except for the case where all distillation training samples have the simple task on, the model can always learn the hard task. This is true for both MLPs and transformers (Figure 6). The rate at which this task is learned is strongly affected by the fraction of samples with only the hard task present. The more hard tasks, the more training time is required to learn to a specific accuracy threshold for a fixed fraction of spurious mechanism. Adding a Jacobian loss term speeds up the learning of harder mechanisms present in both teacher and student datasets. This can be seen by comparing Figure 6(a) to (b): the student reaches higher evaluation accuracy within a fixed number of steps on the dataset with only hard task corresponding to the label.

## 5   Discussion

For all results in this section, the effect of changes such as adding loss terms will differ depending on the modality and dataset, hence the results here should not be considered general.

We observe simplicity bias in the base distillation accuracy plot in 3, where all mechanisms containing spurious latents have higher maximum accuracy scores. Full results (Appendix D, E) show that the presence of the box mechanism in both teacher and student datasets will transfer the box mechanism to the student to near 100% accuracy, to the detriment of learning the image.

Jacobian matching loss on vision datasets has an effect of improved matching of the teacher mechanism, and reduced learning of newer mechanisms only present in the student. In general the results for this loss function are subtle, though the most statistically significant (2 standard deviations minimum)

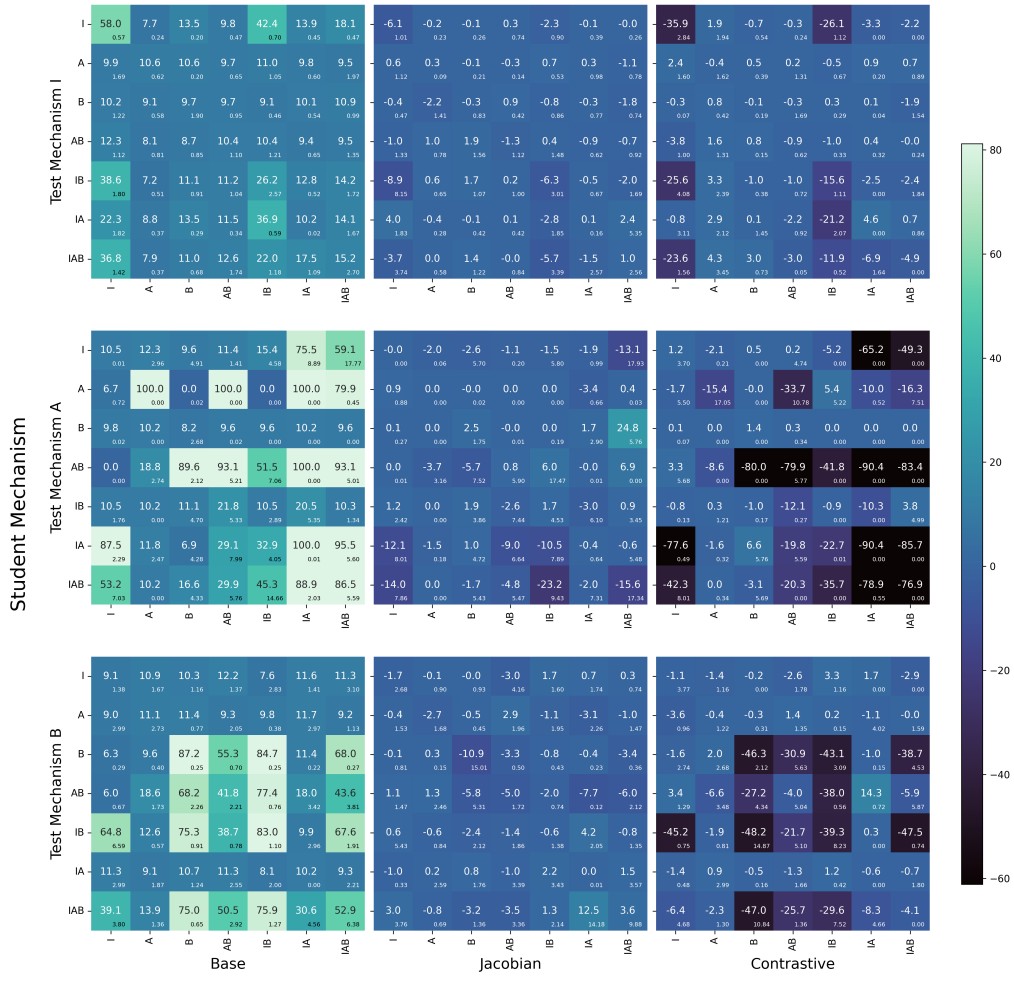

Figure 5: **Final accuracy/accuracy change and standard deviation on various test mechanisms, with dominoes dataset.** Row labels within the heatmaps indicate student mechanism. The base distillation column gives raw values. The Jacobian and contrastive loss columns are differences, given by by new loss function minus base distillation. Column 2: the effect of Jacobian loss is subtle. It typically results in the greatest reduction in performance when the student dataset alone contains the test mechanism. Column 3: contrastive loss leads to reduction in transfer of the spurious mechanisms A and B (rows 2, 3) when both are present in the student and teacher datasets.

.

differences can be found for student and teacher datasets with little overlap (Figure 9 and Appendices D, E). This could be explained by the theory presented in Appendix A. We postulate that due to the complexity and subtlety of this effect, other methods such as matching input-activation Jacobians or using different datasets may produce more pronounced or qualitatively different results.

Across all teacher-student-test dataset triplets we tested, contrastive loss has the lowest teacher-student KL divergence, despite being the slowest to train. This effect also holds on the patterned box dataset, where not only the location but also the pixel values correspond to the label (Appendix E). However, there is an accuracy penalty of around 40% (Appendix D Table 1). This trade-off may be worthwhile in cases where it is important that the nature of the representation the student learns is similar to that of the teacher and large quantities of compute are available. Since prior work [31] shows that with a long enough distillation training, distillation effectiveness increases, there is no reason to believe that increasing epochs will not eliminate this accuracy penalty. We leave exploration of this phenomenon to further work.

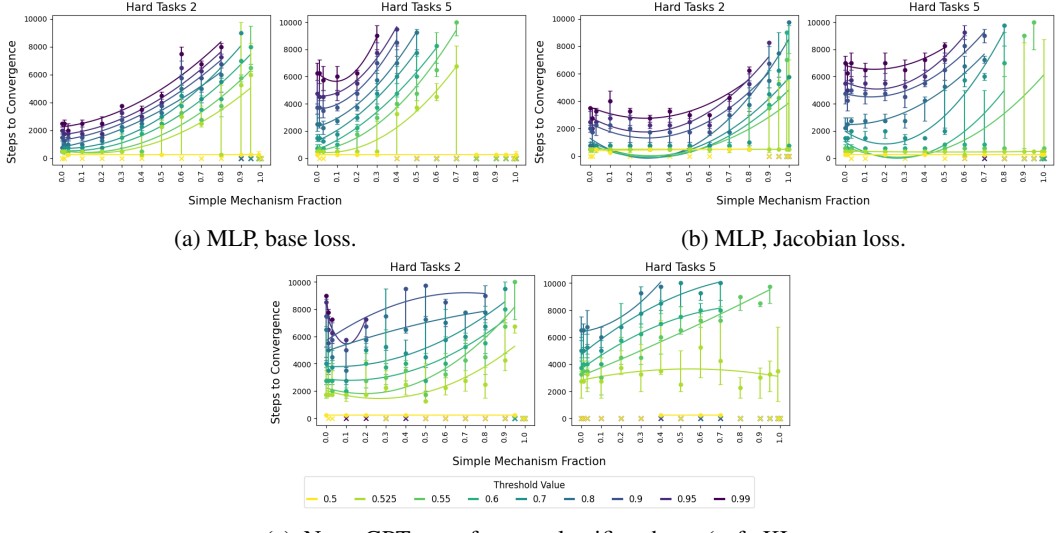

(a) MLP, base loss.

(b) MLP, Jacobian loss.

(c) Nano-GPT transformer classifier, base (soft KL matching) loss.

Figure 6: **Steps to reach particular accuracy threshold on a test dataset vs distillation simple task fraction, parity dataset**. Test dataset: hard task always on, simple task randomised. Distillation dataset: hard task always on, and simple task probability on is given by $x$-axis. Teacher dataset: hard task only (for difference with test dataset, see Appendix B—in this case, teacher and test datasets are identical). When a given accuracy value is never obtained, an 'x' is plotted and the datapoint is omitted from quadratic interpolation. Each data point is a separately trained student. Error bars show steps required in accuracy for $\pm 1$ standard deviation. (a, c) The model always learns the hard task, except when all distillation examples contain the spurious mechanism. Steps to reach a given accuracy for the hard task increases as fraction of simple mechanism in distillation dataset increases. This is expected for per-datapoint simplicity bias. (b) Compare to (a): for a given simple mechanism fraction, fewer training steps are required for reaching a given accuracy threshold.

## 6 Conclusion

In the datasets we examined, we found that Jacobian matching is useful when the teacher dataset is cleaner than the student dataset. Furthermore, we found that contrastive distillation results in a noticeable mitigation of simplicity bias. For both Jacobian and contrastive representation distillation, when the test mechanism either subsets, is a subset of, or only partially overlaps with the teacher and student mechanisms, transfer is reduced when compared to base distillation. In both cases, we also observe slower training. Distillation results are always stopped at a fixed number of epochs, so final accuracy may continue improving in these examples if the student is trained for longer.

Results on the parity dataset also agree with our simplicity bias hypothesis. On distillation datasets where the simple task always corresponds to the hard task's label, the hard task will never be learned by the student. Jacobian matching has the strongest effect on this dataset. It speeds up transfer of the hard task from a clean teacher distilled on a student dataset with the simple task, as long as the dataset has some examples for which the hard task only is predictive of the label.

### 6.1 Further Work

We suggest that further work investigates how exactly the model uses each of the mechanisms, potentially locating 'circuits' corresponding to localized computation of concepts in the network, as per recent interpretability literature [85, 86, 87, 88, 89]. In particular, the mechanism definition may be most useful when each latent dimension corresponds to 'features'—for example, using the Fourier spectrum of image data [90] or gradient spectral clustering [84]. Such human-imperceptible statistical correlations often form the backbone of how models learn algorithms to compute tasks [87, 90], leading to models vulnerable to adversarial attacks. A more modular representation should also allow our data-generating process to align better to how models process information.

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

## Acknowledgements

Thanks go to Herbie Bradley for proofreading and giving pivotal edits for the presentation of the final paper. We would also like to thank Euan Ong and Rudolf Laine for edits and suggestions. Author contributions are as follows: Cindy conducted and evaluated experiments and wrote the codebase for dominoes image experiments and modified Bruno's parity experiment codebase. This paper makes up an extension of her Master's dissertation, a project with David as PI and co-supervised by the co-authors of this paper. Theory and approach was motivated by literature review by Ekdeep, Robert, Cindy and Bruno, with input from David. Bruno wrote the codebase for the parity experiments. The concept of mechanisms and dominoes dataset design were provided by Ekdeep.

## A    Related Work

### A.1    Understanding Distillation.

Several prior works have attempted a study of understanding how knowledge distillation yields highly performing student models, both empirically [39, 41, 31, 44] and theoretically (under strong assumptions) [40, 45, 43, 42]. Most relevant to our work amongst these are the works by Stanton et al. [39] and Beyer et al. [31]. In these papers, the authors evaluate whether the teacher and student model make the same predictions on test samples corresponding to the dataset used for training the student. Their results suggest the teacher and student models relatively rarely make the same incorrect predictions, i.e., they likely focus on different predictive attributes of the data to produce their output. However, due to a lack of formalization, it remains unclear in their work precisely what information did the student acquire from the teacher. We address this limitation by thoroughly controlling for the data-generating process using synthetic datasets. Our analysis enable a precise evaluation of which prediction mechanisms are shared by the teacher and student models and which factors of the training pipeline influence the transfer of these mechanisms.

### A.2    Causal Representation Learning, Disentanglement, and Nonlinear ICA.

Modeling the data-generating process is arguably the foundation of all works on causal and disentangled representation learning [73, 74, 72, 71, 91, 92], as well as the related field of nonlinear-ICA [68, 69, 93, 66, 70]. A highly relevant result from these works is that just because two systems match in their observations (e.g., they generate the same outputs for a given set of inputs), then that does not imply the rules or mechanisms they use for arriving at their outputs are the same [73, 74]. Given the standard distillation pipeline (detailed in Sec. 2) only focuses on matching the student and teacher in the predictions they produce, the results above imply we cannot be certain the models are utilizing the same mechanisms for making their decisions. This raises the question, precisely what mechanisms, if at all, are getting transferred via the distillation pipeline. We address this question in this work.

### A.3    Simplicity Bias in DNNs.

Neural networks have been shown to have an inductive bias towards preferring the simplest hypothesis to explain their data in a supervised setting [57, 58, 59, 60, 35, 94, 95, 96]. While defining simplicity

itself is a difficult problem, the general intuition in these papers is that the number of linear regions required to explain the decision boundary of the model is proportional to the hypothesis's complexity, such that a linear classifier is the simplest hypothesis. Some other work investigating inductive biases in neural networks define simplicity via the general lower-bound complexity measure of the Kolmogorov complexity of the function the NN approximates [97, 98, 78]. Others view it through the lens of effective parameter count or degeneracy of the trained model [76, 99, 100]. As we show, simplicity bias plays a non-trivial role in which mechanisms actually transfer to the student in a distillation pipeline, with more advanced methods essentially improving performance by enabling transfer of more mechanisms.

### A.4 Measuring Similarity of Neural Networks.

As we instantiate it, a transfer of knowledge essentially implies two models produce similar representations or predictions. A few prior works have used this intuition to evaluate similarity of two neural networks as well, defining measures like prediction mismatch [101, 102, 103, 39] and variants of Canonical Correlation Analysis [104, 105, 106, 107]. However, by not accounting for the data-generating process itself, these measures are only capable of measuring how similar two networks are in their outputs on a given distribution, specifically the one the models were trained on. As we show, the notion of a mechanism for prediction must account for the data-generating process and hence measure the model's outputs on out-of-distribution data. A few papers have proposed data-driven methods for measuring similarity of two neural networks [108, 109], however they generally limit their measures to similarity under some differomorphism of the inputs. These measures are highly correlated with predicting a model's performance [110], but are relatively less useful for our purpose of understanding which mechanisms are transferred in a distillation setting, which requires the ability for a more precise characterization.

### A.5 Learning with forgetting, distribution shift and Jacobian matching

This section is key to understanding how the Jacobian loss function may behave under distribution shifts between teacher and student datasets. In particular, while existing theory shows learning with forgetting (LwF) conducts matching on subsets of the teacher's dataset, Srinivas et al. [61] suggest more work is to be done investigating matching between a teacher and a student under *more structured distribution shifts than noise addition.*

Learning with forgetting involves using both hard and soft distillation on smaller target (student) dataset. The pre-trained teacher may be trained on a different, larger source dataset. The teacher can produce noisy and incorrect results on unseen data. For LwF to work well, activations of pre-trained teacher on target dataset must contain information about the source dataset.

Srinivas et al. introduce a formalisation for a general distillation loss bound in learning with forgetting. This becomes relevant in our work for exploring large statistical dataset domain shifts between the teacher (source) dataset, which is often fixed and/or inaccessible in practice, and the distillation dataset (target dataset). They claim LwF approximates distillation on a subset of the teacher's source dataset:

**Proposition A.1.** *Suppose $f(\cdot)$ is the untrained student network, and $g(\cdot)$ is the pre-trained teacher network. Let $\boldsymbol{x}, \boldsymbol{y}$ be the input image and ground truth label, and $|\mathcal{D}|$ the dataset size. Let the distillation loss function be $\ell(f(\boldsymbol{x}), g(\boldsymbol{x}))$. Also assume Lipschitz continuity for the loss function with constant K, and a valid distance metric in $\psi_{\boldsymbol{x}}$ on the input vector space. This metric can be most usefully thought of as distance in feature or embedding space.*

$$\|\ell(\boldsymbol{x}_1) - \ell(\boldsymbol{x}_2)\| \leq K\psi_{\boldsymbol{x}}(\boldsymbol{x}_1, \boldsymbol{x}_2)$$

*Define the asymmetric Haussdorf distance between two sets in the input space.*

$$\mathcal{H}_a(A, B) = \sup_{a \in A}(\inf_{b \in B} \psi_{\boldsymbol{x}}(a, b))$$

*Then, the following should hold, where the left hand side is the loss on the teacher's dataset, and the right hand side is the maximum loss on the student's dataset.*

$$\frac{1}{|\mathcal{D}_l|} \sum_{x \sim \mathcal{D}_l} \ell(f(\boldsymbol{x}), g(\boldsymbol{x})) \leq \max_{\boldsymbol{x} \sim \mathcal{D}_s} \ell(f(\boldsymbol{x}), g(\boldsymbol{x})) + K\mathcal{H}_a(\mathcal{D}_l, \mathcal{D}_s) \tag{5}$$

*The Haussdorf distance limits the maximum bound on the loss on the teacher's dataset. By restricting $\mathcal{D}_s$ to the overlap with $\mathcal{D}_s$, the Haussdorf distance is minimised.*

This work claims that if there is no overlap, then distillation will not be of benefit, since the Haussdorf distance depends on the overlap in the two datasets.

## B    Details of Training and Hyperparameters

**Dominoes Dataset**    We use three mechanisms—a spurious box replacing certain pixels of a CIFAR-10 image and an Fashion-MNIST image concatenated on top of a CIFAR-10 image, making a $64 \times 32$ domino. Where the mechanism is not present, it is replaced by black pixels (F-MNIST mechanism) or CIFAR-10 pixels (box mechanism). The box mechanism is an $8 \times 8$ patch of random pixels. Appendices D, E also details results on a box mechanism where the pattern within the box corresponded to the label. Here, the model can either learn the box's location or the pixel placement within them. This was chosen to test whether our results transferred between qualitatively different types of spurious features and relative simplicity of mechanisms affect conclusions drawn.

**Parity Dataset**    This dataset assigns a label for the parity of a a subset of a Boolean string. Which subset or specific hard task is used depends on the control bits at the start of the string. There is also a subset of the string dedicated to the simple task. This is designed to be easier to learn because the substring is shorter than each hard task substring, and its location within the string is always fixed. A probability of simple task of 0.5 means that there is a 50% chance that the parity of the simple task bits matches that of the true label (parity of 'hard task' bits).

**Models**    All experiments use self-distillation to remove confounding issues with architecture size for transfer capability.

For the dominoes dataset, we use PyTorch's pretrained ResNet18, with weights reset using PyTorch's module default parameter reset for fully connected layers (uniform with no bias), and Kaiming initialisation for convolutional layers. Adaptive average pooling with output size (1,1) was used (global pooling).

For the parity experiments, we use a 3-hidden layer MLP with 50 neurons each. The transformer is a nano-GPT implementation with a classifier built in the final layer. It has 3 layers, 4 heads per layer and an embedding dimension of 256.

**Training hyperparameters**    A hyperparameter sweep was conducted for Jacobian and contrastive loss to softmax KL loss ratio, which minimised evaluation loss. The values chosen were 0.15 Jacobian and 0.03 for contrastive loss.

For the dominoes dataset with PyTorch's ResNet18, a batch size of 64, SGD with no momentum or dropout, and cosine LR scheduler were used. The number of evaluation batches was 50, and training was done for 20,000 iterations. Initial and final learning rates were 0.01 and 0.001. A base distillation temperature of 30 was chosen after initial hyperparameter search.

For the parity dataset, a batch size of 256, AdamW with no momentum, cosine LR schedule or dropout and gradient clipping of 0.25 was used. The learning rate was fixed at 0.001. Hard task bits was 8 in total, while number of bits per hard task was 3. Simple task bits was 2. A distillation temperature of 1 was used throughout. Evaluation batches was 10, and training was done for 10,000 iterations.

**Jacobian matching loss implementation**    It is possible to approximate this computationally expensive loss function with Jacobian terms with largest magnitude, but we need to compute the full Jacobian for this. A heuristic is the output variable involving correct answers or largest output logits (often corresponds to right label in trained models). We implement the full Jacobian matching as well as an optional top-k class Jacobian matching for large datasets such as CIFAR-100, where top-1

matching may not capture sufficient information. We only investigate input-output Jacobian, though code is available for feature-matching Jacobians as in Srinivas et al. [61].

**Contrastive representation loss implementation**    We use the version of code available publicly from Tian et al. [67]. This was edited to remove the memory buffer. In practice, a large number of negative pairs should be seen each time the loss is calculated to better approximate the lower bound, so the authors use a memory buffer is used to store previously-used feature representations of each datapoint. However, the error difference between N=64 and N=4096 is around 0.8%. This minor difference meant our implementation used only images present in each training batch and no memory buffer, and does not make use of their code for the memory buffer. An embedding is used to transform the dimensionality of the student and teacher representations onto the same space and normalise them. A dot product similarity is then used, with a softmax-like transformation to normalise as a probability $P(C = 1|T, S)$. In their original implementation, both the student and teacher are used as the anchor. When the student is the anchor, its output for a specific image are compared against multiple feature samples drawn from the teacher's outputs. Assuming the teacher is fixed (offline distillation) means we only use the student as an anchor. Finally, the contrastive representation requires intermediate layer representations. In all experiments, the representation was lifted from the 8th residual block's second convolutional layer post batch-norm (or the input to the final average pool). This hopefully captures higher-level representations about the data. Other layers were not attempted in this project, though it is possible that the qualitative result differs depending on the layer and architecture.

**Training Protocol and Experimental Design**    In distillation, the student and teacher see the same samples and the student may be able to learn a randomised mechanism. This can happen if there is independence of representation by the teacher for hard or simple tasks, or the teacher was trained on a dataset with this spurious mechanism not randomised with respect to the labels. For this reason, we remove the spurious mechanism during training and replace the data with zeros, where possible. This was done with the image dataset, since in the parity dataset, a binary string of zeros still corresponds to a valid parity label. The parity dataset contains multiple hard tasks, treated as one semantically important mechanism. With 2 latent variables, the number of different datasets is rather low at 3. This dataset is not so useful for distillation shift, but instead used to investigate basic NLP setups and changing fraction of spurious mechanism in the dataset.

In typical counterfactual evaluations, randomising the mechanisms which we do not want to be learned should suffice. This is the approach taken for the parity dataset. The parity dataset uses two types of evaluation datasets.

- Test datasets: the simple task either has a 50% probability of matching parity with the hard task, or always matches the hard task. We flip the last bit of the simple task if it does not match the hard task. These datasets make up the results in Figure 6.
- Counterfactual datasets: randomize latent bits of the mechanisms involved, and randomize the labels in a matching manner. For the hard task, the function mapping latents to observed variables is the identity map—for standalone evaluation, this randomisation does nothing. However, the point of the counterfactual datasets is to compare performance to the 'factual' (training) datasets. For the simple task, the latents versus observed datapoints differ only in the last simple bit. This bit is flipped based on a Bernoulli distribution with parameter set by the probability of simple task. These datasets make up the results for entropy and top-1 mismatch probability in Appendix F.

## C   Teacher Results

Figures 7, 8 show performance for the dominoes dataset on test test datasets during training for early-stopped teachers. The purpose of these plots is to show a baseline in relative ease of learning different mechanisms via counterfactual evaluations, before distillation is used. The following trends are noted from the teacher training data.

Observation 1: no clear modularity. A clean assumption is if the model can learn tasks 1 and 2 together, then it has formed separate circuitry which localises information within the model for each of the tasks. Based on performance of the teacher model for the image corrupted with a box (Figure 7, IA), neither sub-mechanism is learned in isolation: the sum of the image and box accuracies is not 100%. Further work may like to test what modifications can be universally made for images to yield

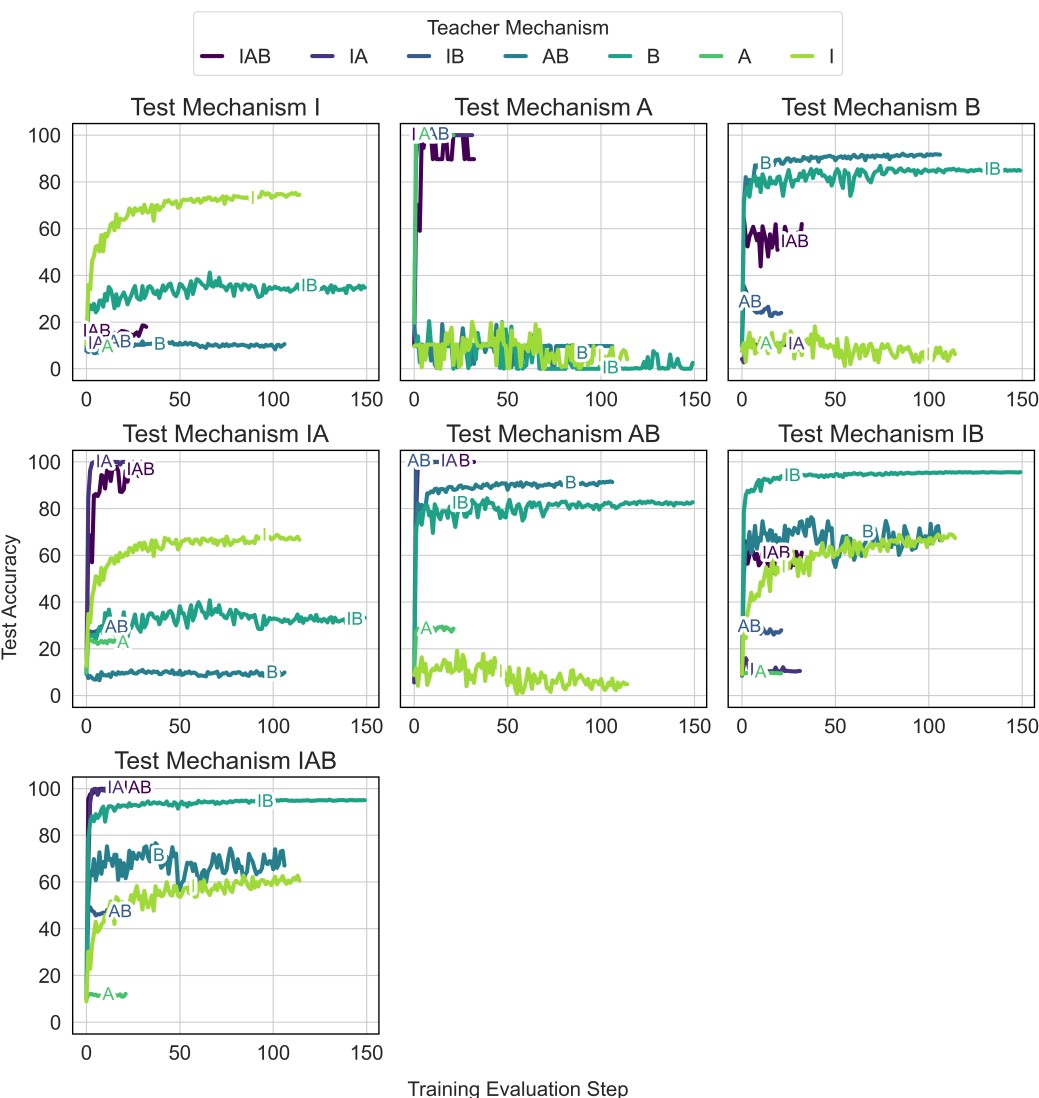

Figure 7: Teacher accuracy over training time with dominoes dataset, random box mechanism.

more modularity. For example, if the box mechanism size decreases or a constant pixel value is used, will mechanism IA will become more modular? This is possible, if the model is less incentivised to use edge detection. Aligning our latent discrete variable constituents (our with features a model sees when the overall task (IAB) is fed in helps with producing results where performance on the independent tasks helps predict what happens when shifts occur without necessarily having to have trained the model at hand on these datasets.

Observation 2: static information is likely preferred. Note how simplicity bias is shown for teachers trained on A (box mechanism) and tested on box or datasets with the box mechanism plus one other mechanism (IA, AB). This behaviour does not carry through for the test dataset with all 3 mechanisms (IAB). Assumptions about what might make for a good toy dataset are difficult to know a-priori. While the box mechanism is expected to be simplest, Figure 7 shows F-MNIST explains most of the performance on mechanism AB. This has implications for what can be considered a mechanism in more naturalistic data settings. Preliminary results not shown here (code available in codebase (TOLINK)) show using a smaller box with static information increases the box mechanism's relative simplicity.

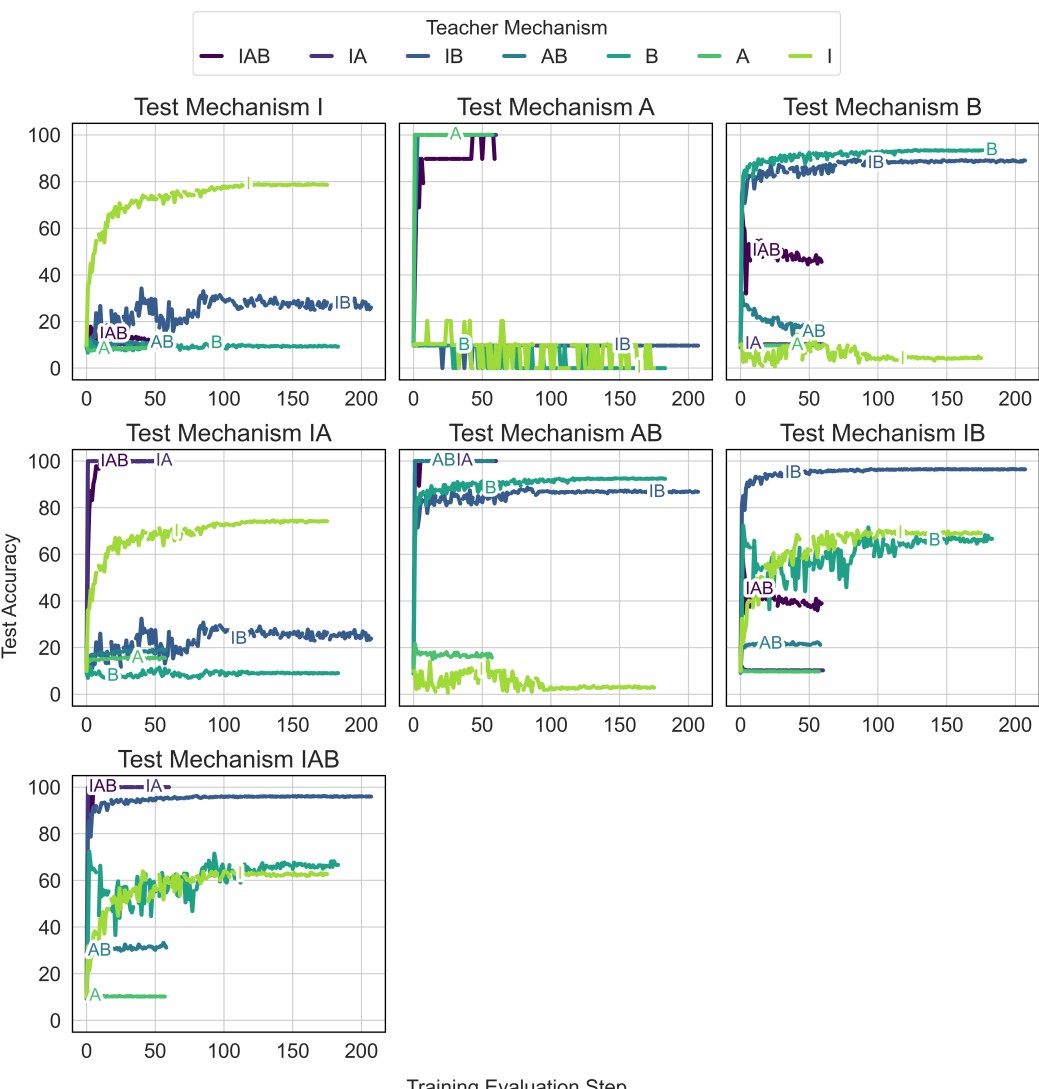

Figure 8: Teacher accuracy over training time with dominoes dataset, label-specific box pattern.

## D   Dominoes Dataset Final Accuracy

In all graphs in this section and succeeding sections with the dominoes dataset, the shorthand I (image), A (spurious mechanism 1 box), B (spurious mechanism 2 F-MNIST) will be used when referring to the mechanisms present in the datasets. Columns within the heatmaps hold results when trained on datasets with the same teacher mechanism, while rows all have students trained with the same distillation dataset mechanism ('student mechanism'). We report final test accuracy, with mean and population standard deviation estimate from 3 seeds. The values for base distillation are raw values, while those for Jacobian and contrastive are values obtained with the new loss function minus values obtained for base distillation.

### D.1   Random Box

The table below outlines a more fine-grained grouping of all results, conditional on a greater than 4 percentage point difference with base distillation loss. While this threshold is somewhat arbitrary, it captures any values that are at least the sum of each value's standard deviation away from each other. Data was drawn from all 343 teacher, student and test mechanism combinations. For simplicity, results are to nearest percent difference. Results to 1 decimal place are in the heatmaps below.

**Notation.** $IS$ (in S): the test mechanism is equal to/a subset of $S$. $NT$ (not teacher): the test mechanism has no overlap with $T$. $SM$ (student more): the test mechanism is present in both student and teacher datasets, but $S$ contains extra mechanisms which are not in $T$. $EQ$ (equal): $S, T$ and test mechanisms are identical. $SS\ S, T$ (subset student and teacher): the entire test mechanism is a subset of both student and teacher mechanisms. This subset does not need to be the same for the student and teacher mechanisms. $PT$ (part teacher): there exists a test mechanism subset and $T$ subset which matches. For row headings, $J$ = Jacobian, $C$ = contrastive.

| | IS | | IT | | IS, IT | | | P or S, T | | | |
|---|---|---|---|---|---|---|---|---|---|---|---|
| | NT | PT | NS | PS | SM | TM | EQ | PS, IT | PT, NS | PS, NT | SS, S, T |
| J | −9 | −8 | +6 | +6 | −2 | +0 | −6 | −12 | −13 | −10 | −9 |
| C | −24 | −42 | −28 | −32 | −37 | −48 | −40 | −25 | −15 | −38 | −49 |

## D.2 Patterned Box

Now we consider the box mechanism where the pattern within each patch corresponds to a label-specific crop of a cartoon two-tone Mandelbrot set. The information per class is static.

As in Table 1, the table below outlines a more fine-grained grouping of all results, conditional on a greater than 4 percentage point difference with base distillation and to nearest percentage.

Table 2: Grouped final accuracy change compared to base distillation for label-specific pattern box.

**Notation.** $IS$ (in S): test mechanism equal to/subset of $S$. $NT$ (not teacher): test mechanism has no overlap with $T$. $SM$ (student more): test mechanism present in both student and teacher datasets, but $S$ contains extra mechanisms which are not in $T$. $EQ$ (equal): student, teacher and test mechanisms are exactly equivalent. $SS\ S, T$ (subset student and teacher): entire test mechanism is a subset of both student and teacher datasets. However, this subset does not need to be the same for student and teacher. $PT$ (part teacher): there exists a test mechanism subset and $T$ subset which matches. For row headings, $J$ = Jacobian, $C$ = contrastive.

| | IS | | IT | | IS, IT | | | P or S, T | | | |
|---|---|---|---|---|---|---|---|---|---|---|---|
| | NT | PT | NS | PS | SM | TM | EQ | PS, IT | PT, NS | PS, NT | SS, S, T |
| J | −1 | +2 | NA | −11 | +4 | −15 | NA | −5 | −5 | −7 | −15 |
| C | −4 | −22 | +6 | −7 | −3 | −22 | −41 | −11 | +7 | −14 | −23 |

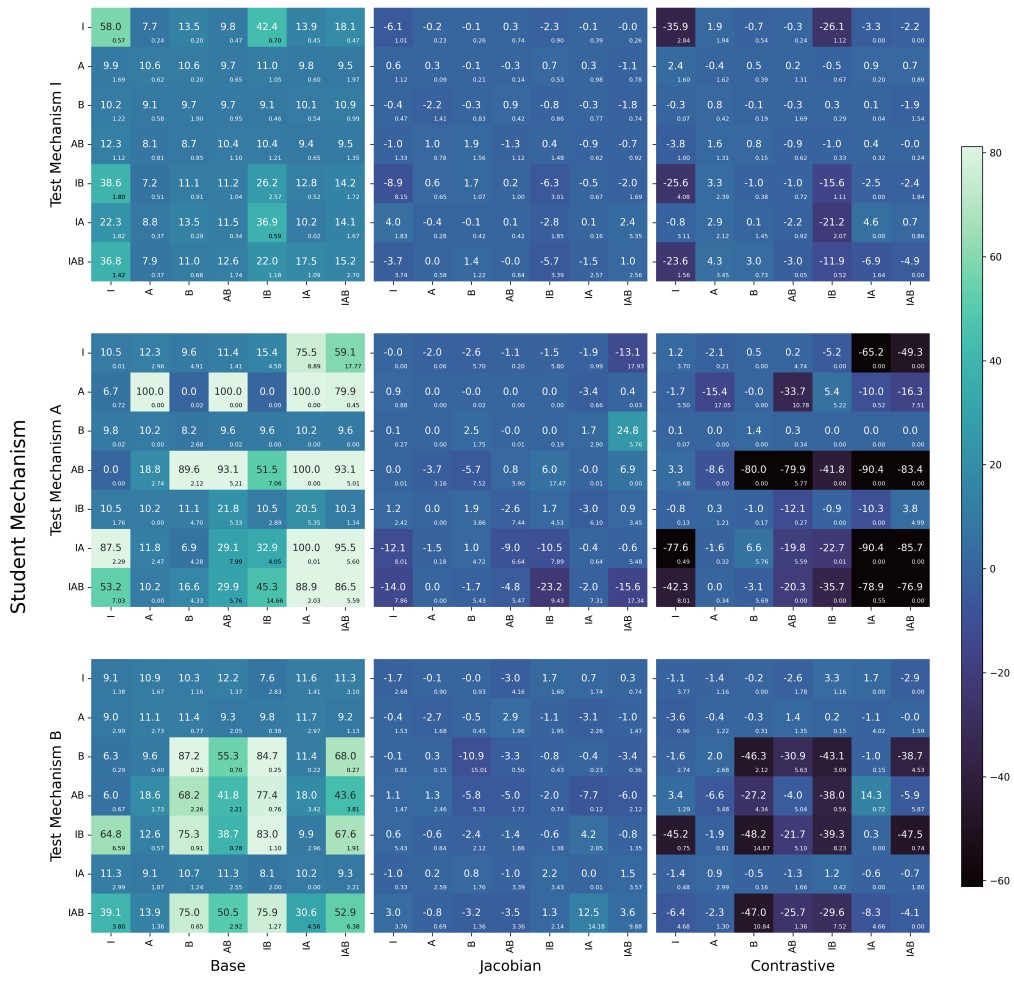

Figure 9: Final accuracy and standard deviation, random pixel box mechanism dominoes dataset.

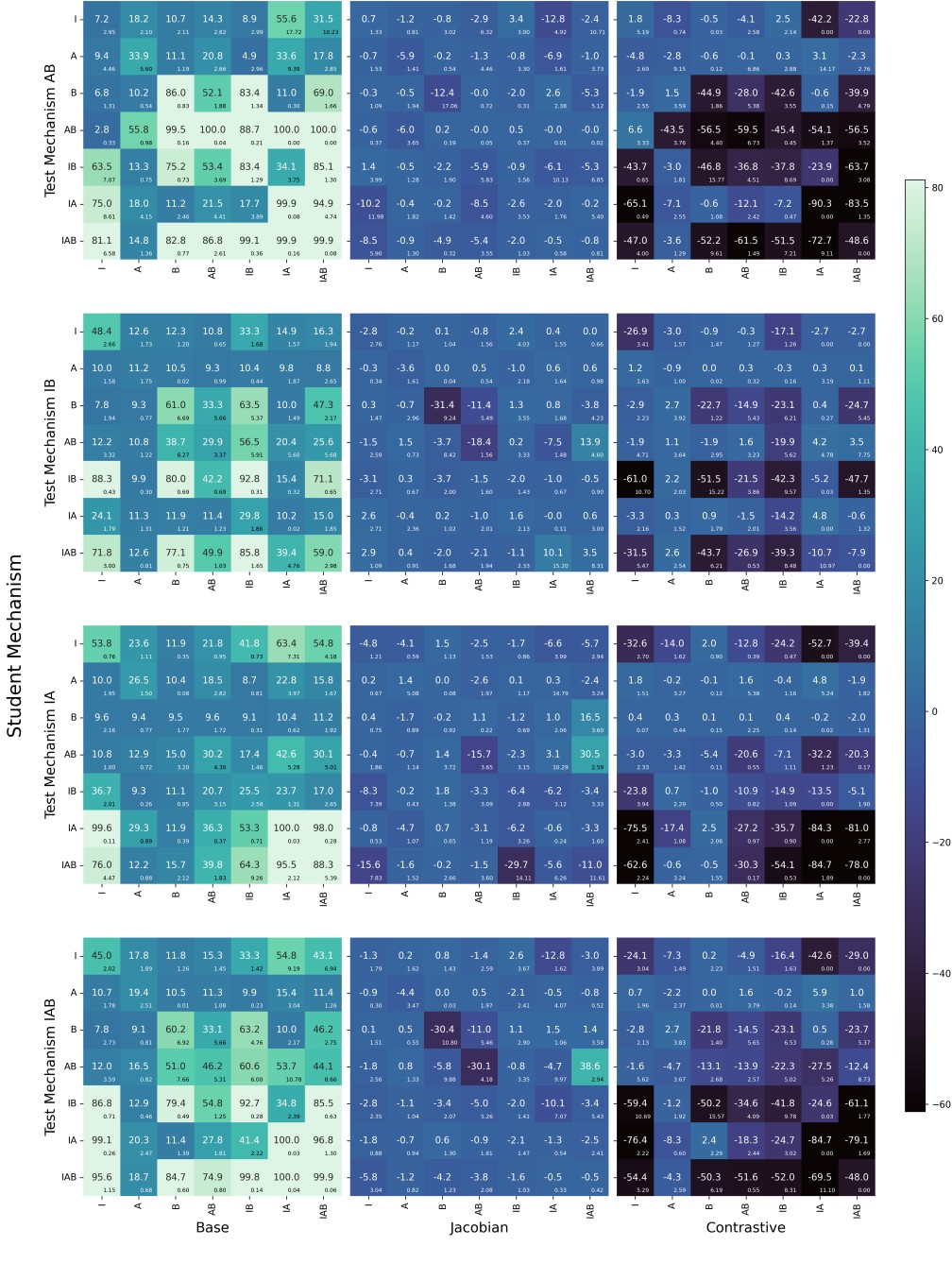

Figure 10: Final accuracy and standard deviation, random pixel box mechanism dominoes dataset.

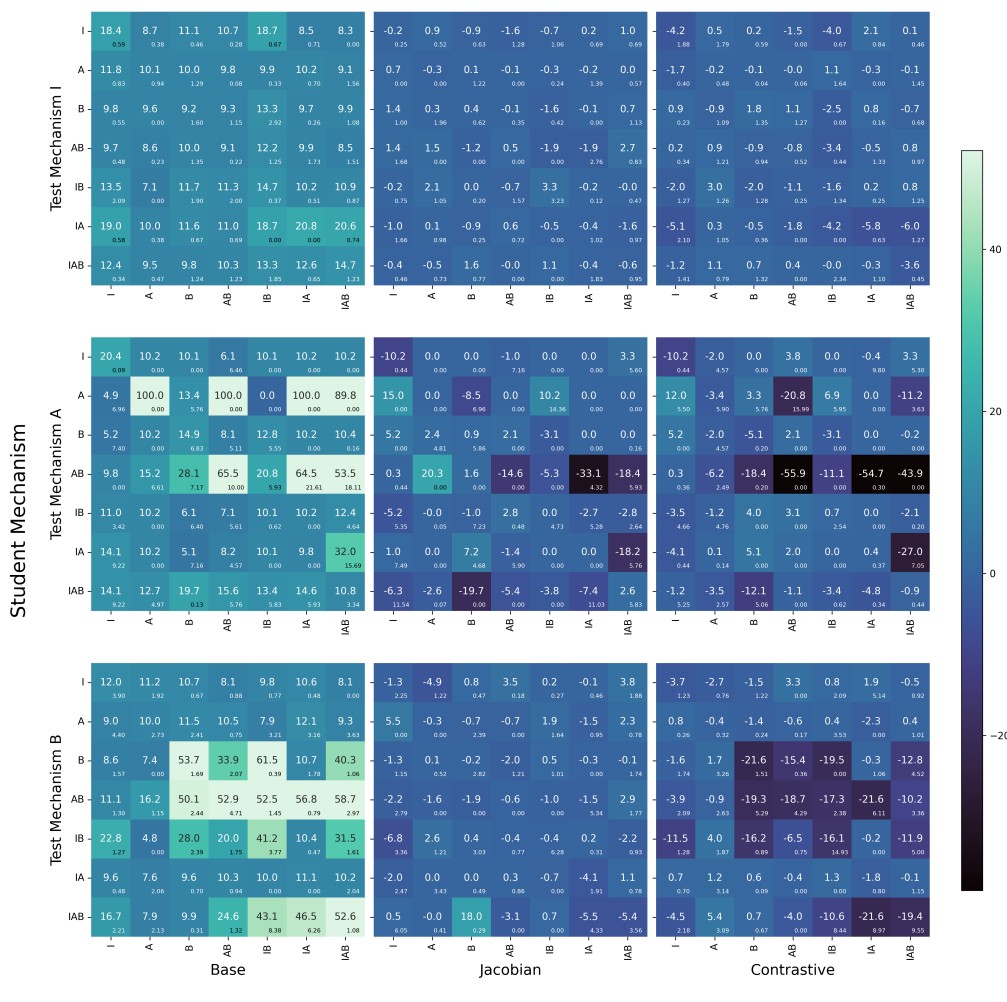

Figure 11: **Final accuracy and standard deviation**, on label-specific box pattern dominoes dataset.

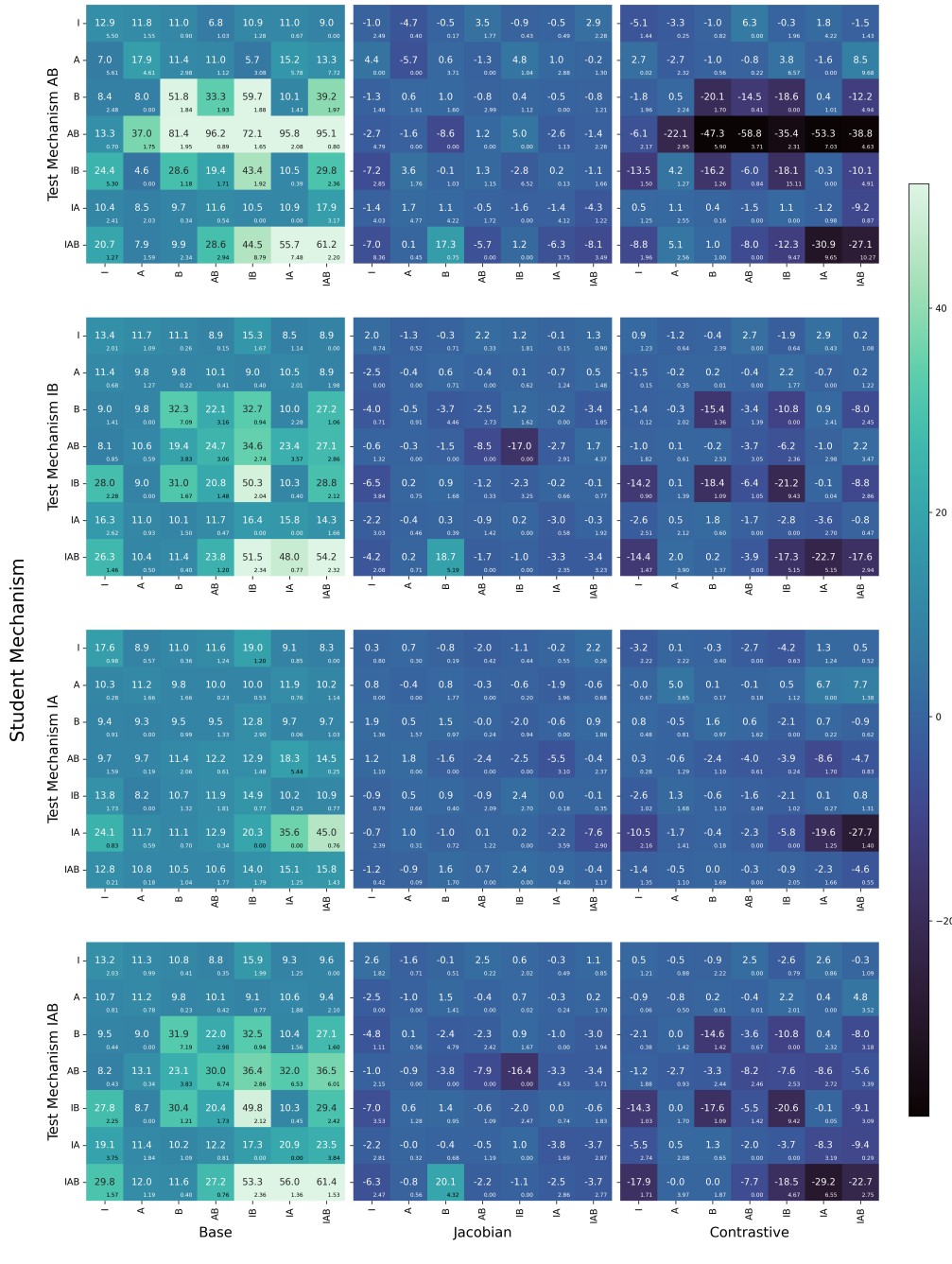

Figure 12: **Final accuracy and standard deviation**, on label-specific box pattern dominoes dataset.

# E   Dominoes Dataset KL Divergence

For the final KL divergence values in this section, there exist many cases where the KL divergence can increase (i.e. worse matching of teacher and student distributions), but accuracy stays the same or increases. Similarly, there are cases where the KL divergence can decrease, while evaluation final accuracy decreases. In all heatmaps in this section, the base distillation column gives raw mean final values. The Jacobian and contrastive loss columns are mean differences, given by value obtained using new loss function minus value obtained with base distillation.

## E.1   Random Box

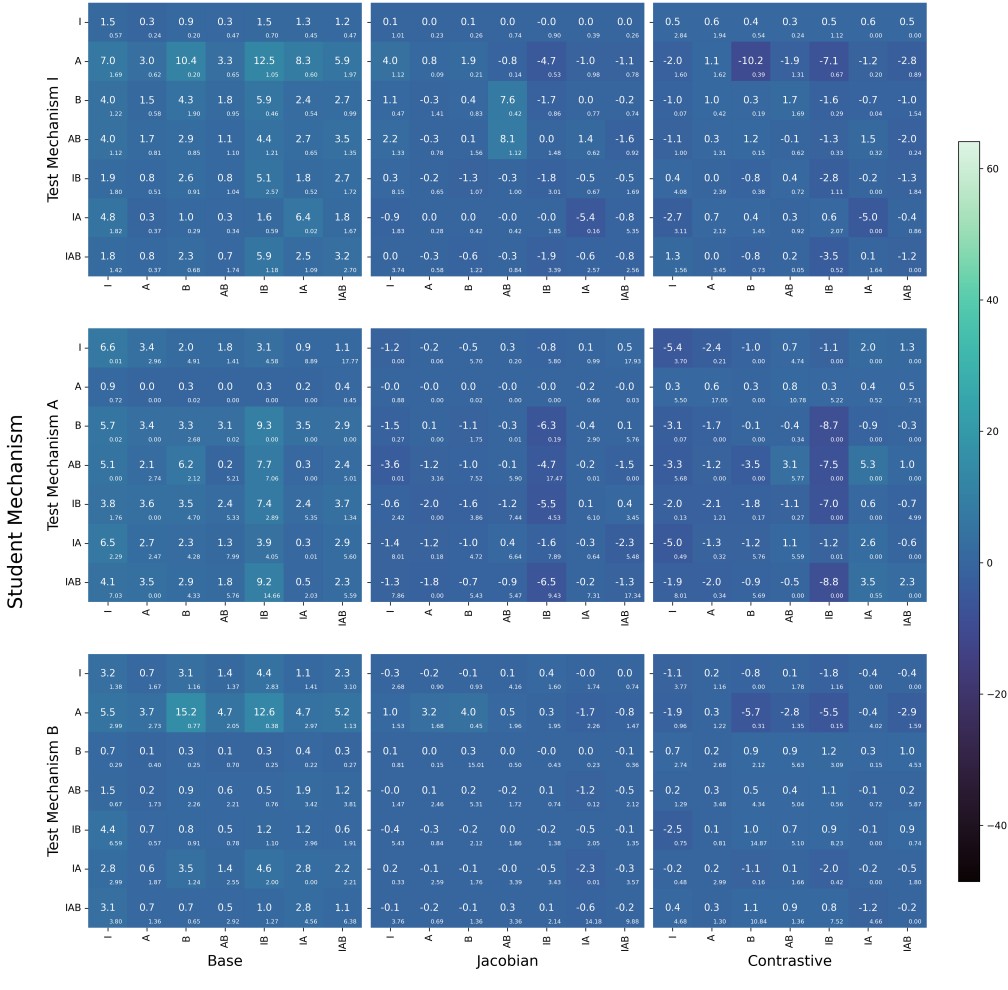

Figure 13: **Teacher to student KL divergence** $KL(T|S)$ **and standard deviation**, on random pixel box mechanism dominoes dataset.

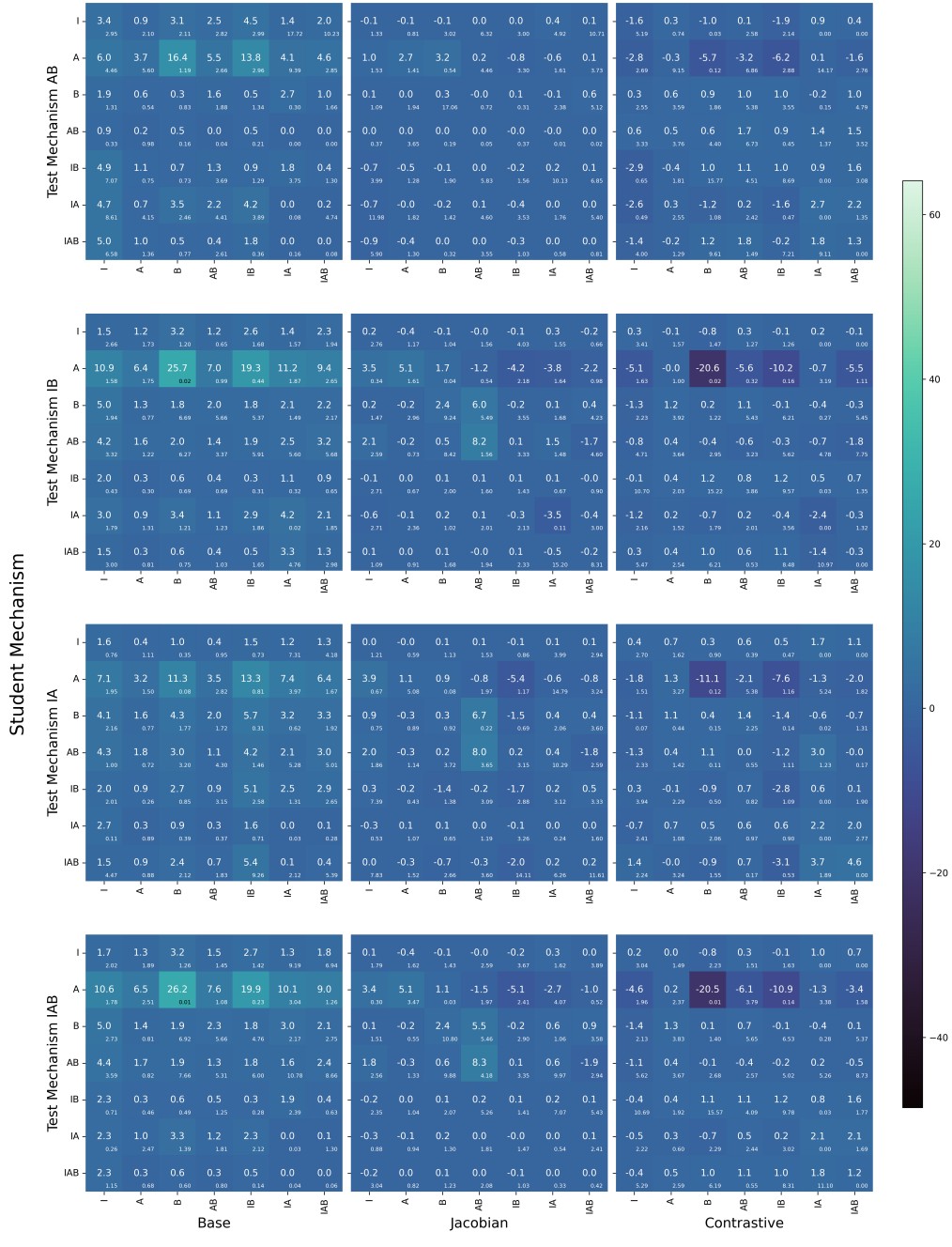

Figure 14: **Teacher to student KL divergence** $KL(T|S)$ **and standard deviation**, on random pixel box mechanism dominoes dataset.

## E.2    Patterned Box

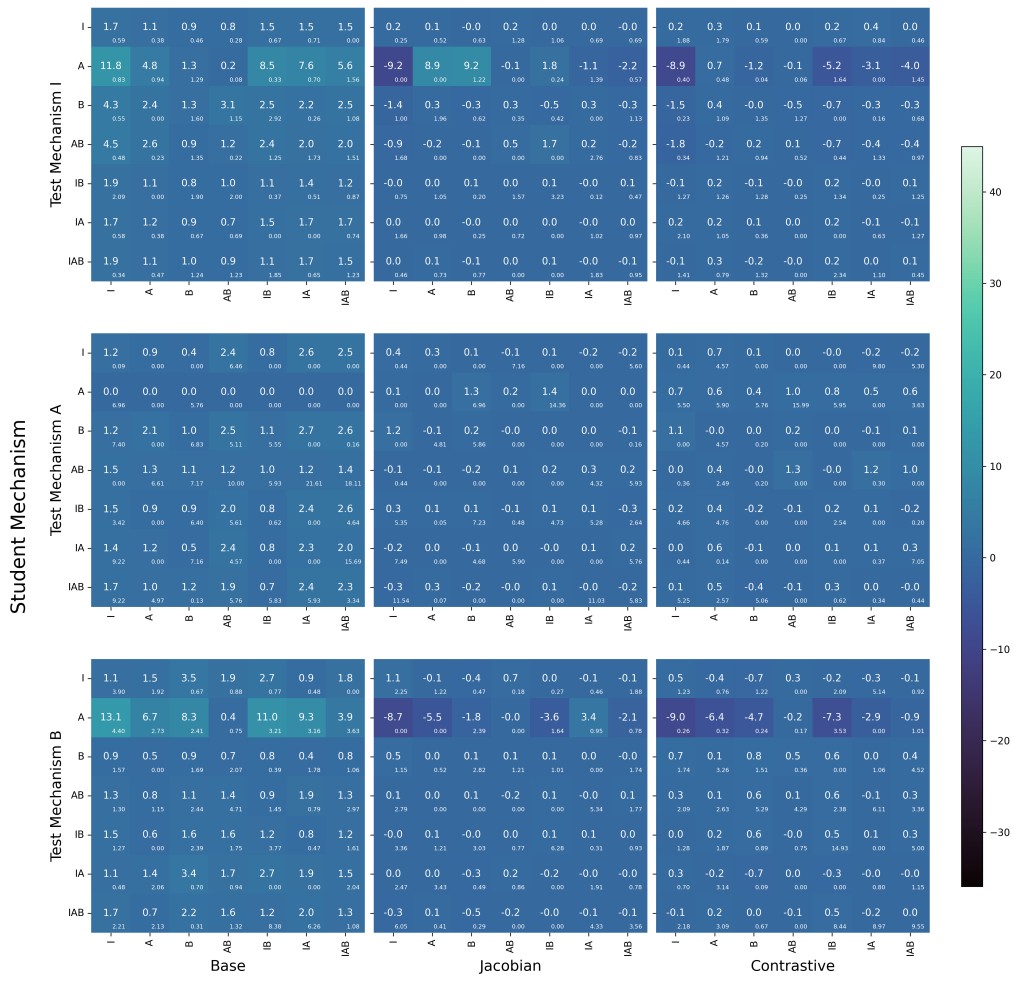

Figure 15: **Teacher to student KL divergence** $KL(T|S)$ **and standard deviation** on label-specific box mechanism dominoes dataset.

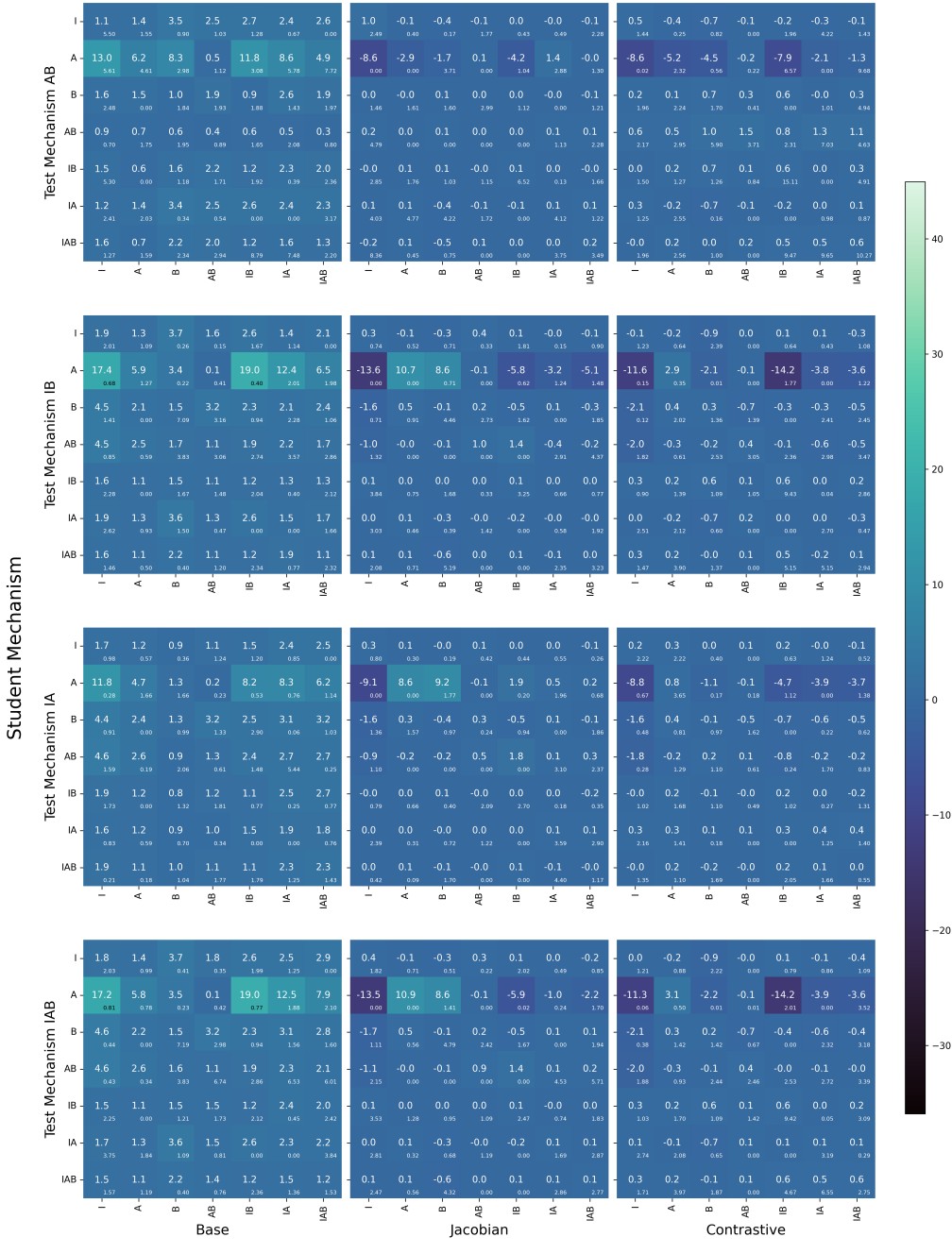

Figure 16: **Teacher to student KL divergence** $KL(T|S)$ **and standard deviation** on label-specific box mechanism dominoes dataset.

# F    Parity Spurious Fraction Results

Figure 17 summarizes accuracy over training time for the MLP and transformer using Jacobian or base distillation loss, on number of hard tasks ranging from 3 to 7. It is an extension of figures in Section 4.3. For each run, we also report prediction dissimilarity probability and entropy change on test datasets. This is detailed in Subsection F.1.

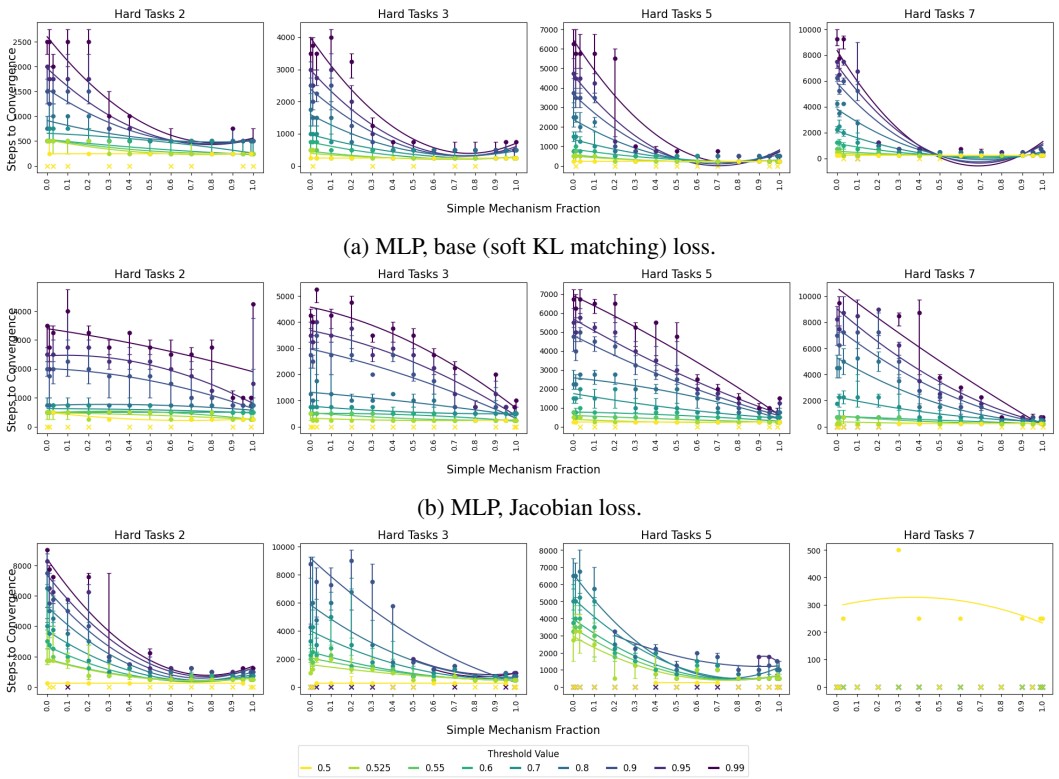

Figure 17: **Steps to reach particular accuracy threshold vs distillation simple task fraction.** Test dataset: simple task always on, hard task always on. Distillation dataset: hard task always on, and simple task probability on is given by $x$-axis. The teacher dataset has hard task on only. the x-axes varies the probability of simple task corresponding to correct label in the distillation dataset. Each data point is a separately trained student. Training was done for 10000 iterations and 5 seeds. The teacher dataset has hard task bits predictive of parity, and simple task bits are randomised. The threshold accuracy curves compress full training evolution curves, giving steps to achieve a given accuracy. If that accuracy value is never obtained, an 'x' is plotted and that datapoint is omitted from interpolation.

## F.1    Counterfactual Evaluations

The following metrics are reported in this section. For simplicity, the concept of the hard or simple task's parity matching the label in any dataset example is colloquially called 'on/off'.

- Maximum prediction change: probability of disagreement of top predicted label between student evaluated on the training dataset examples, and examples where a) either the hard task or simple task is randomised (depending on which test dataset is used), b) it is as specified during training (base example). There are $M$ type (a) examples for each type (b) example.
- Maximum prediction entropy: entropy of the unique predictions of randomised examples. This is computed for a fixed number $M$ times for each base example.

If the randomised examples lead to a large entropy increase, then it can be assumed that the task that was randomised was important for the model to be certain about its predictions. If this is accompanied by a significant probability of top-1 disagreement, it is likely the model was relying on this mechanism for its predictions.

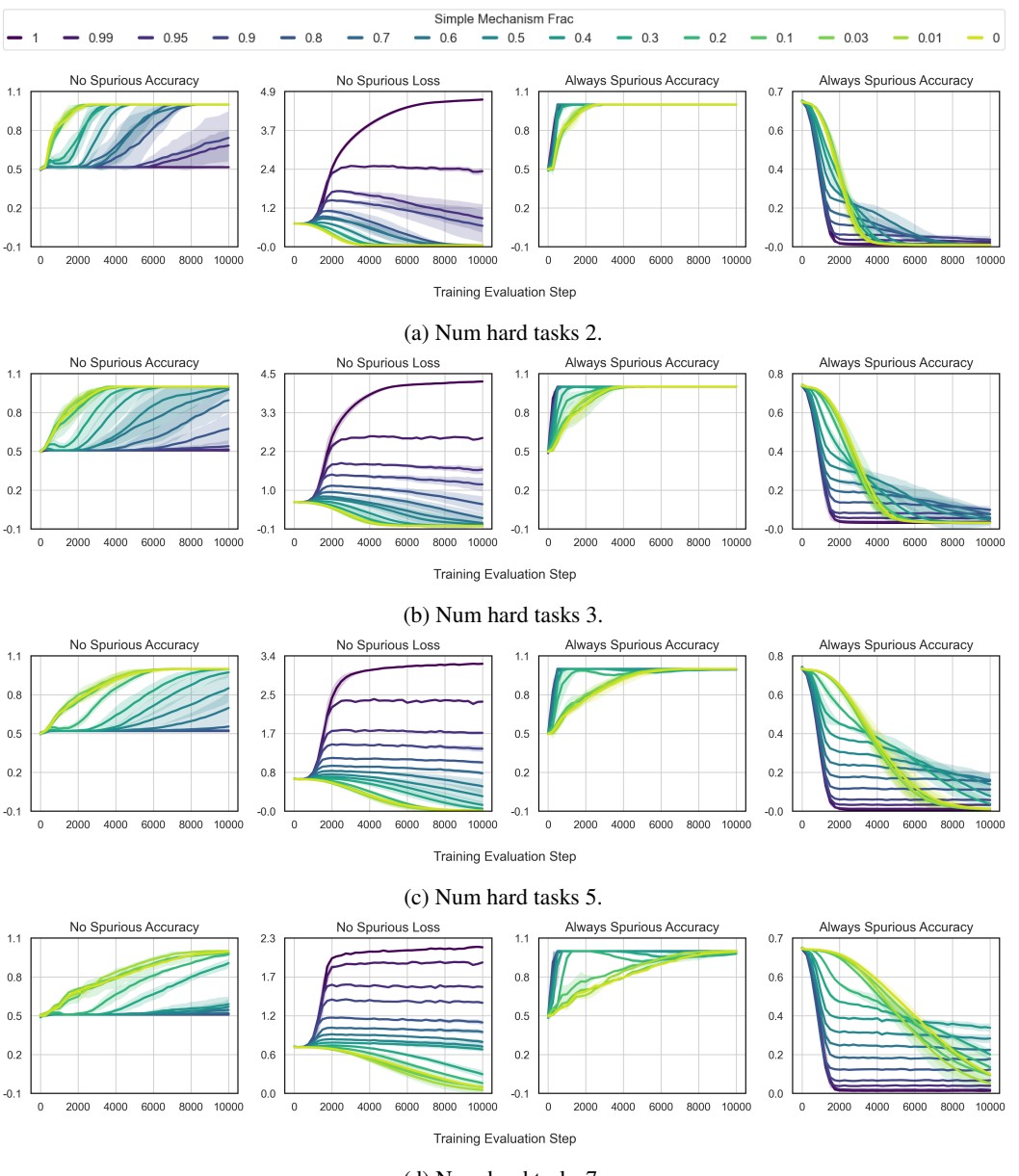

Figure 18: **Accuracy and loss**. Test datasets: simple task on/off, hard task always on. MLP, base loss, teacher dataset hard task on only. Distillation dataset has varying probability of simple task on (legend), and hard task always on.

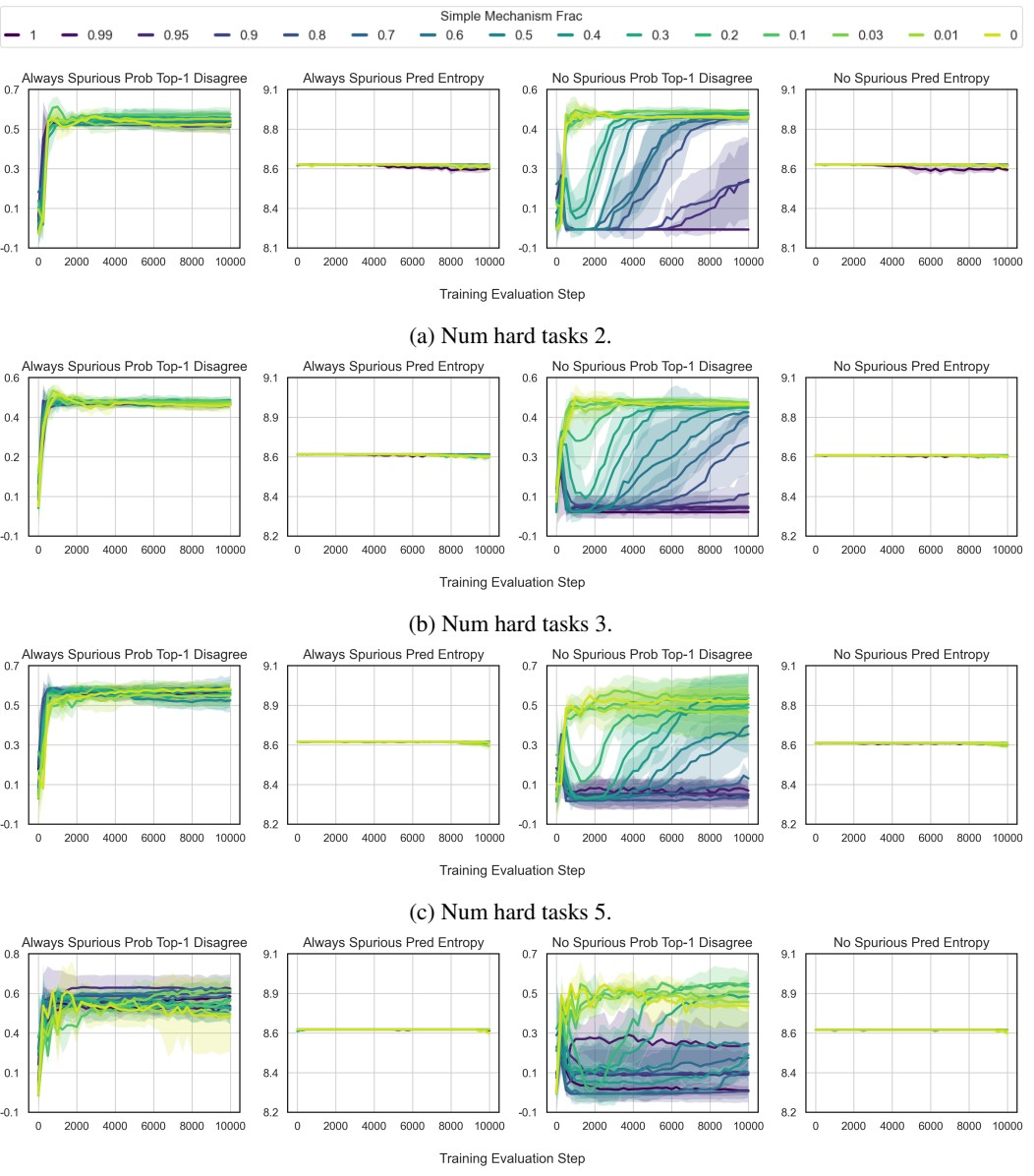

(a) Num hard tasks 2.

(b) Num hard tasks 3.

(c) Num hard tasks 5.

(d) Num hard tasks 7.

Figure 19: **Mean top-1 disagreement probability and entropy**. Counterfactual datasets: simple task on/off, hard task randomised. MLP, base loss, teacher dataset hard task on only. The distillation dataset has varying probability of simple task on (legend), and hard task always on. For the no spurious delta max prediction column, as expected, results match those of the test dataset with no spurious feature during training (Figure 18 column 1). This is because counterfactual and test examples should be equivalent between these two datasets.

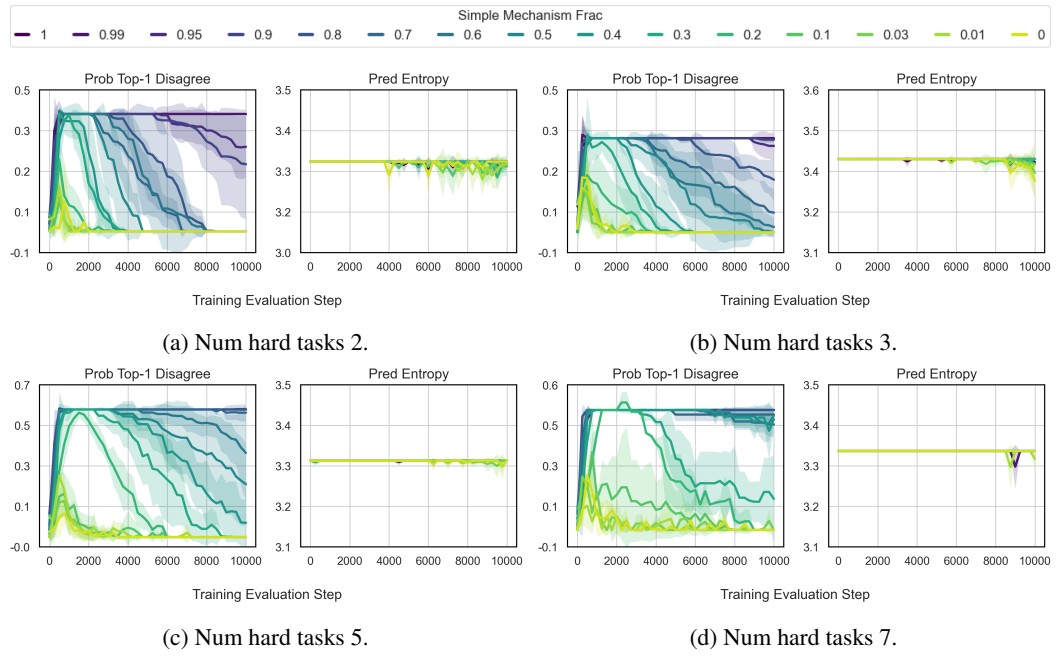

(a) Num hard tasks 2.

(b) Num hard tasks 3.

(c) Num hard tasks 5.

(d) Num hard tasks 7.

Figure 20: **Mean top-1 disagreement probability and entropy**. Counterfactual datasets: simple task randomised. MLP, base loss, teacher dataset hard task on only. The distillation dataset has varying probability of simple task on (legend), and hard task always on.

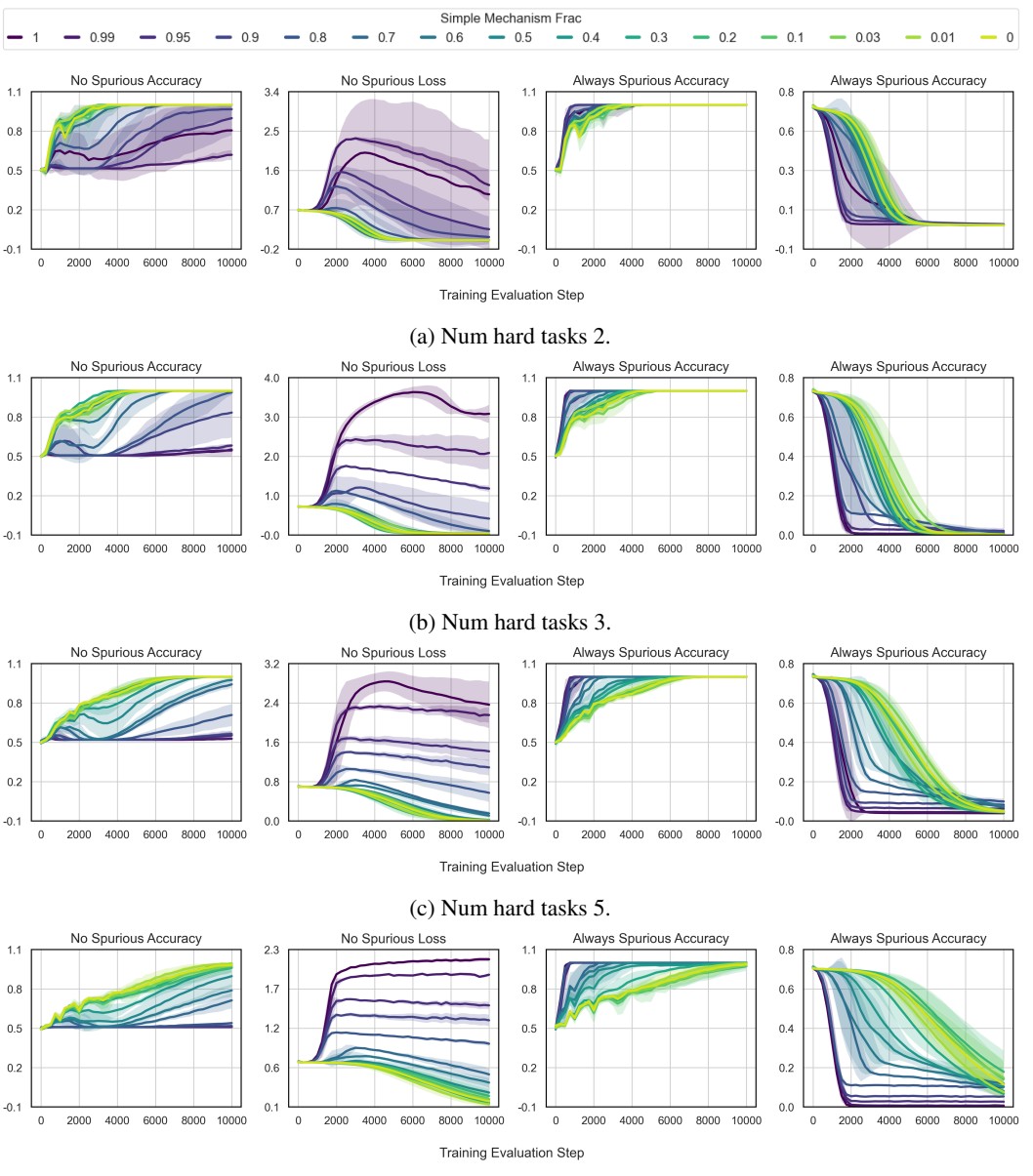

(a) Num hard tasks 2.

(b) Num hard tasks 3.

(c) Num hard tasks 5.

(d) Num hard tasks 7.

Figure 21: **Accuracy and loss**. Test datasets: simple task on/off, hard task always on. MLP, Jacobian loss, teacher dataset hard task on only. Distillation dataset has varying probability of simple task on (legend), and hard task always on.

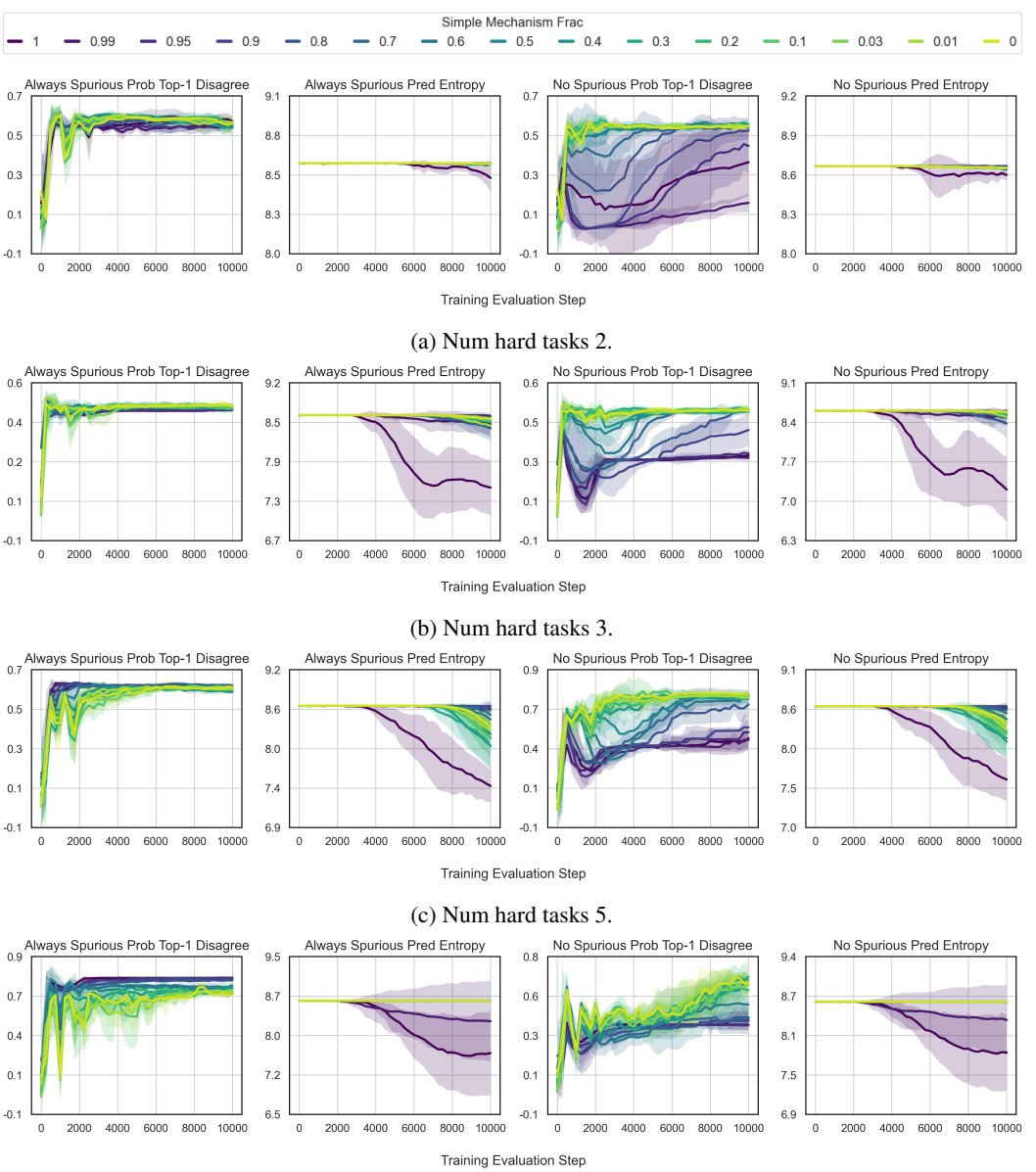

Figure 22: **Mean top-1 disagreement probability and entropy**. Counterfactual datasets: simple task on/off, hard task randomised. MLP, Jacobian loss, teacher dataset hard task on only. The distillation dataset has varying probability of simple task on (legend), and hard task always on.

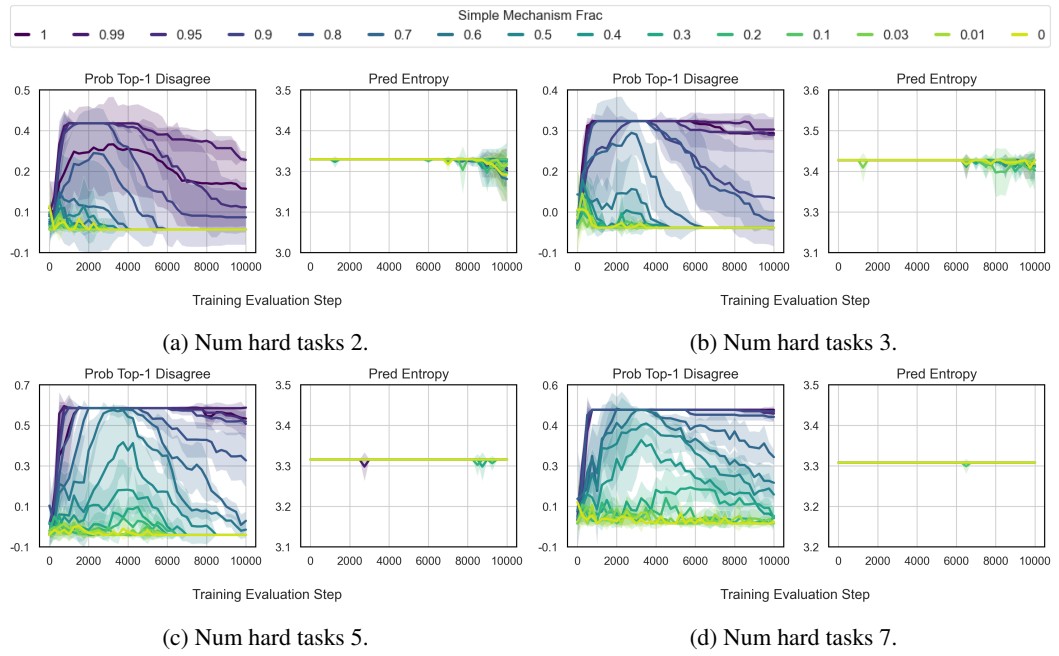

(a) Num hard tasks 2.

(b) Num hard tasks 3.

(c) Num hard tasks 5.

(d) Num hard tasks 7.

Figure 23: **Mean top-1 disagreement probability and entropy**. Counterfactual datasets: simple task randomised. MLP, Jacobian loss, teacher dataset hard task on only. The distillation dataset has varying probability of simple task on (legend), and hard task always on.

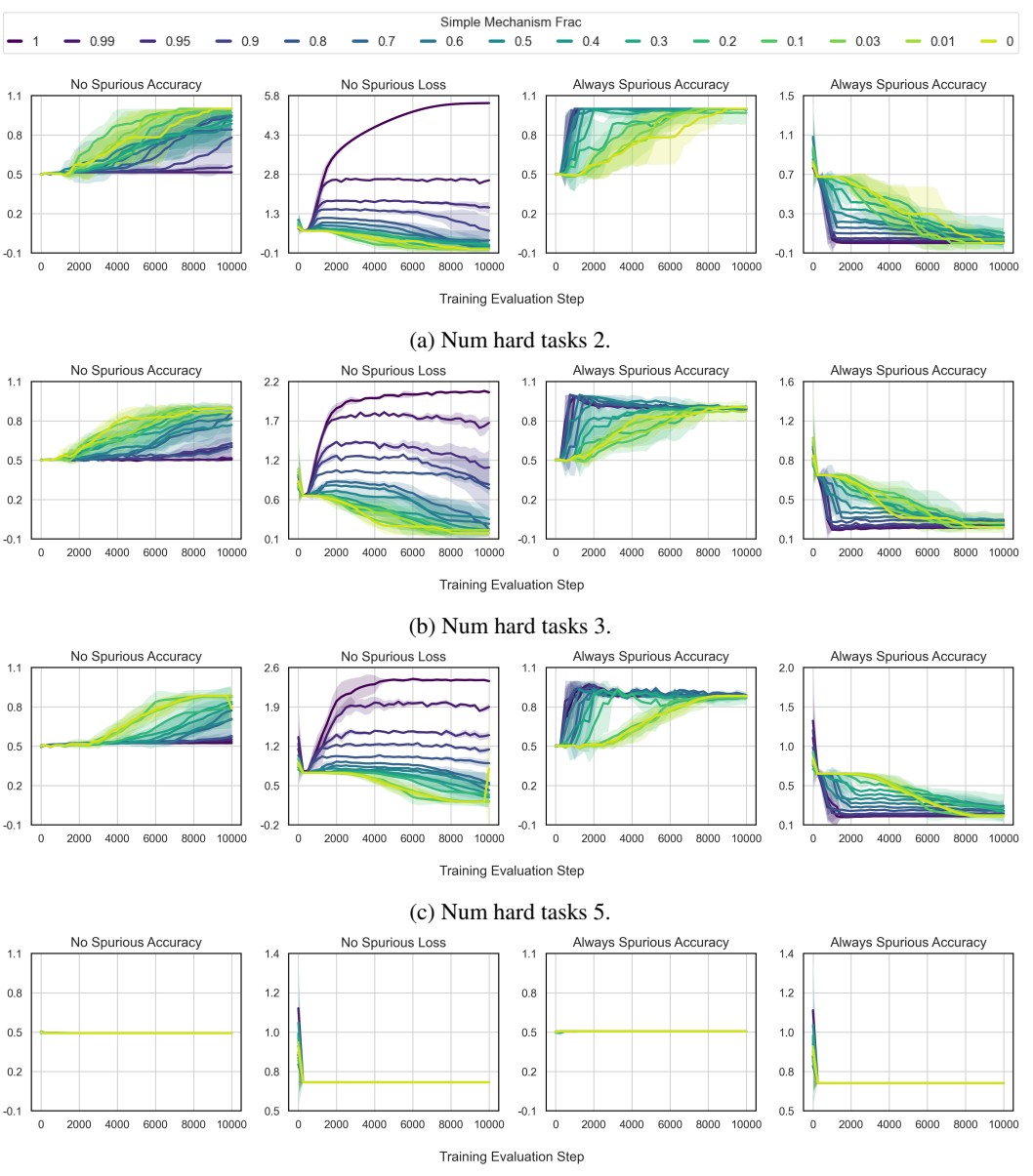

(a) Num hard tasks 2.

(b) Num hard tasks 3.

(c) Num hard tasks 5.

(d) Num hard tasks 7.

Figure 24: **Accuracy and loss**. Test datasets: simple task on/off, hard task always on. Transformer classifier, base loss, teacher dataset hard task on only. Distillation dataset has varying probability of simple task on (legend), and hard task always on.

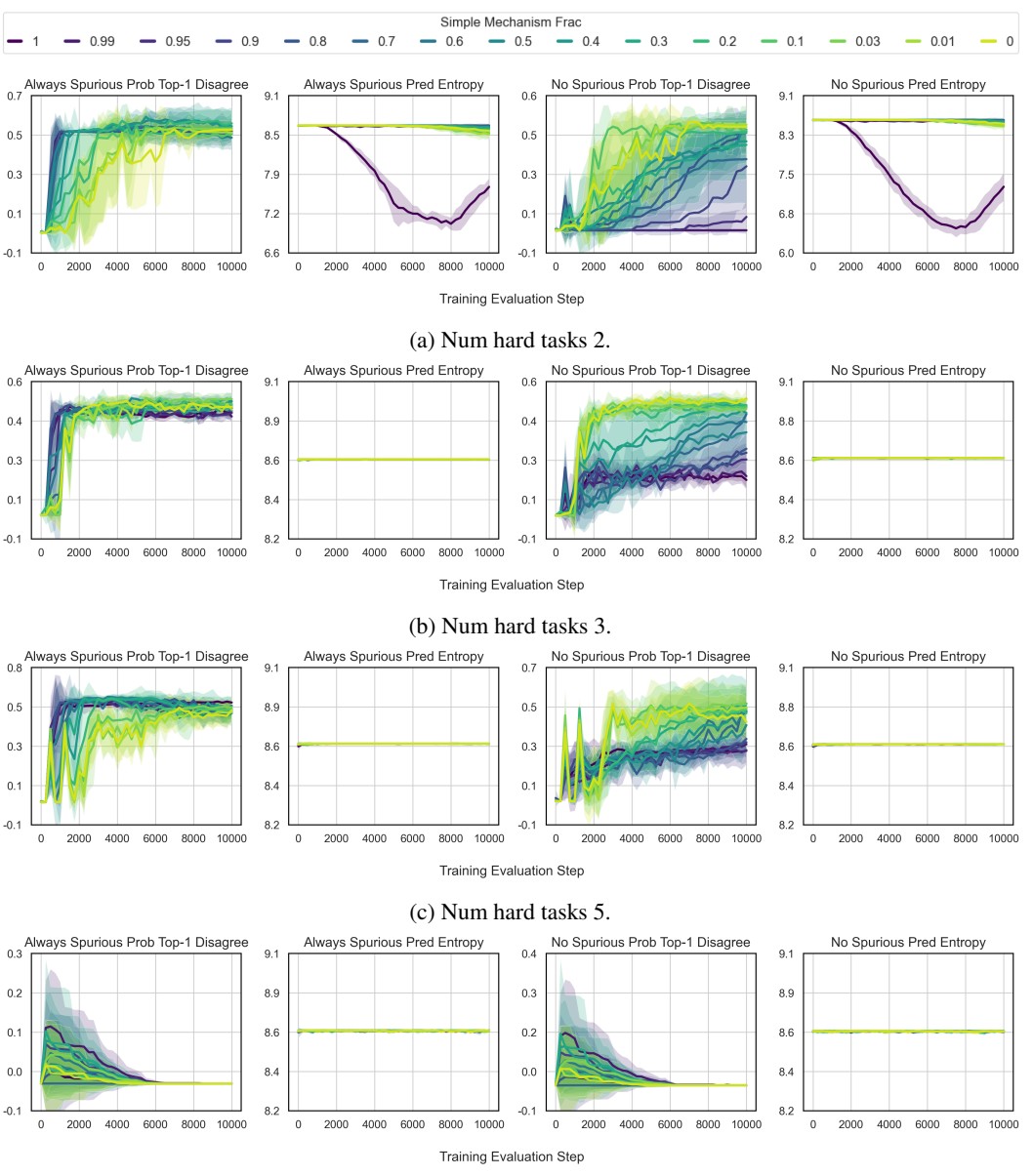

(a) Num hard tasks 2.

(b) Num hard tasks 3.

(c) Num hard tasks 5.

(d) Num hard tasks 7.

Figure 25: **Mean top-1 disagreement probability and entropy**. Counterfactual datasets: simple task on/off, hard task randomised. Transformer classifier, base loss, teacher dataset hard task on only. The distillation dataset has varying probability of simple task on (legend), and hard task always on.

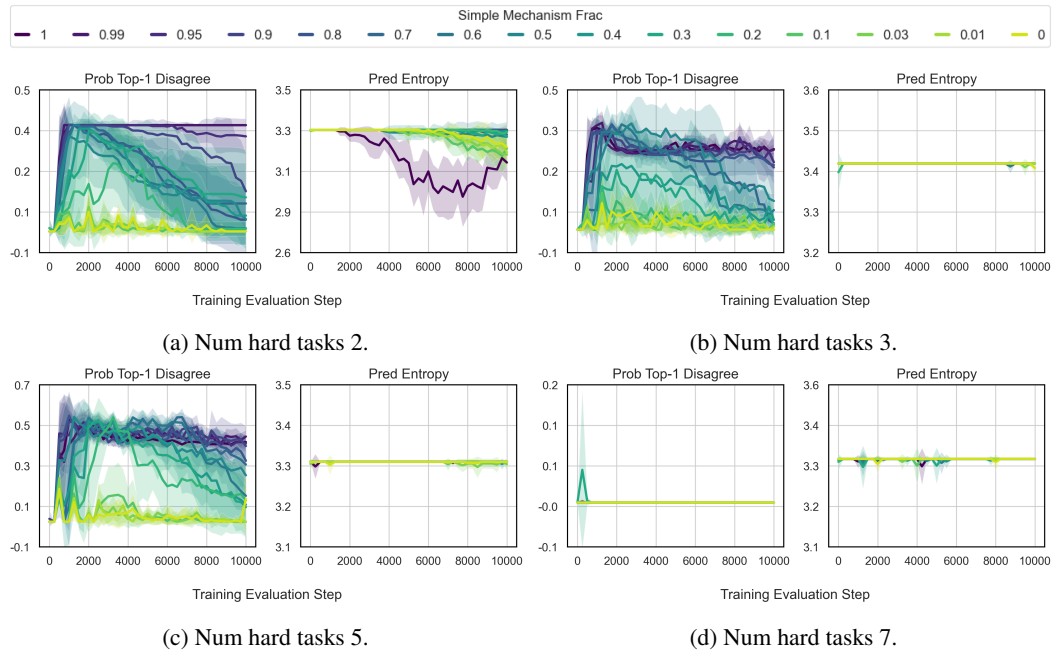

Figure 26: **Mean top-1 disagreement probability and entropy**. Counterfactual datasets: simple task randomised. Transformer classifier, MLP loss, teacher dataset hard task on only. The distillation dataset has varying probability of simple task on (legend), and hard task always on.

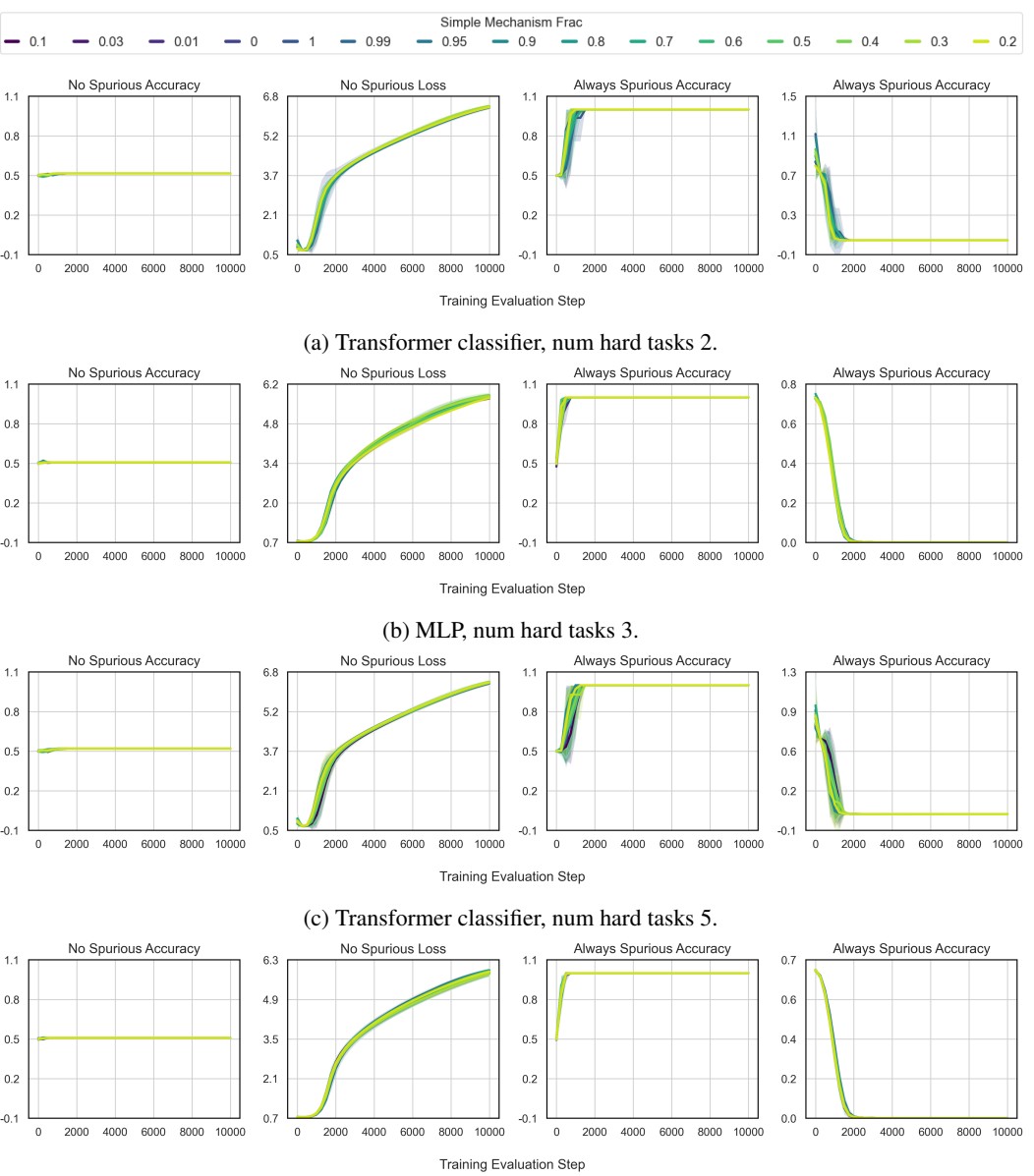

(a) Transformer classifier, num hard tasks 2.

(b) MLP, num hard tasks 3.

(c) Transformer classifier, num hard tasks 5.

(d) MLP, num hard tasks 7.

Figure 27: **Accuracy and loss**. Test datasets: simple task on/off, hard task always on. Base loss, teacher dataset hard task and simple task on. Distillation dataset has varying probability of simple task on (legend), and hard task always on.

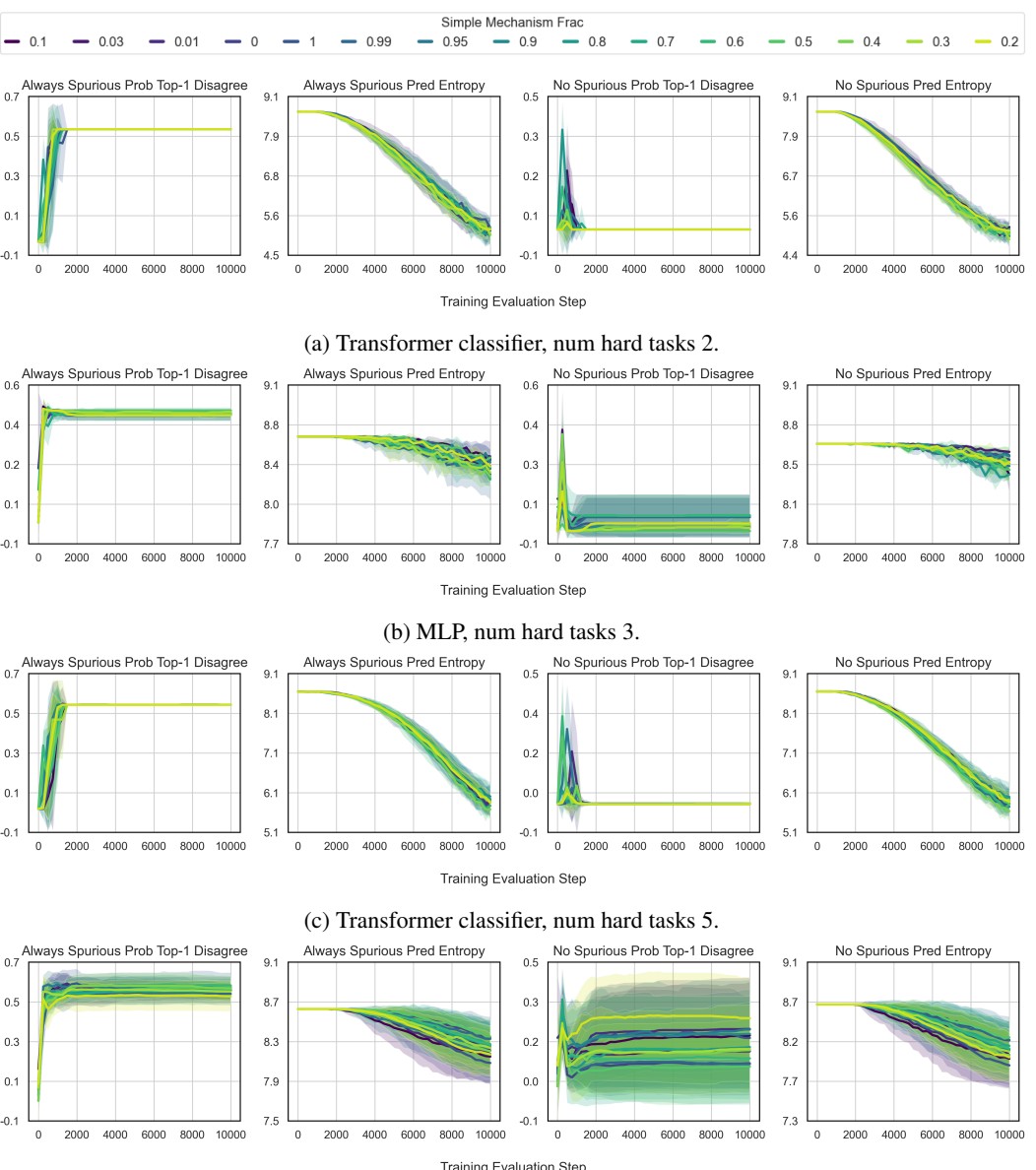

(a) Transformer classifier, num hard tasks 2.

(b) MLP, num hard tasks 3.

(c) Transformer classifier, num hard tasks 5.

(d) MLP, num hard tasks 7.

Figure 28: **Mean top-1 disagreement probability and entropy**. Counterfactual datasets: simple task on/off, hard task randomised. Base loss, teacher dataset hard task and simple task on. The distillation dataset has varying probability of simple task on (legend), and hard task always on.

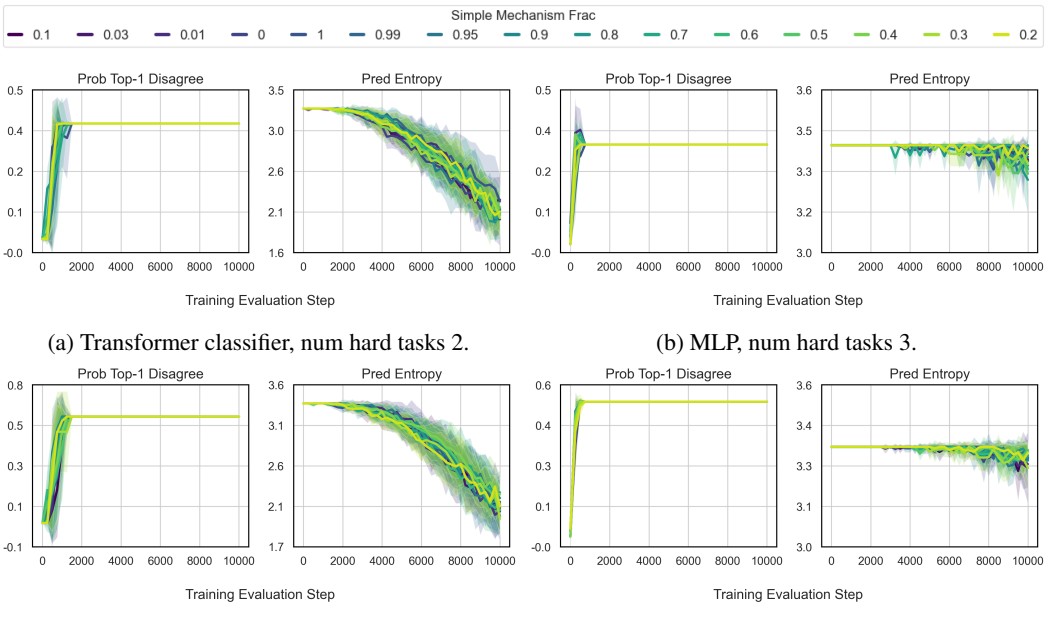

(a) Transformer classifier, num hard tasks 2.

(b) MLP, num hard tasks 3.

(c) Transformer classifier, num hard tasks 5.

(d) MLP, num hard tasks 7.

Figure 29: **Mean top-1 disagreement probability and entropy**. Counterfactual datasets: simple task randomised. Base loss, teacher dataset hard task and simple task on. The distillation dataset has varying probability of simple task on (legend), and hard task always on.

