# OpenReview forum: "What Mechanisms Does Knowledge Distillation Distill?"
_NeurIPS.cc/2023/Workshop/UniReps — UniReps Poster_

### Official Review · Reviewer_4DxD · 2023-10-23
**Understanding Mechanism Sharing in Knowledge Distillation**

**Rating:** 7
**Confidence:** 3

**Review:**

**Introduction**

This paper dives into the nuances of knowledge distillation, focusing on understanding and formalizing the concept of 'knowledge' and its transfer from larger teacher models to smaller student models. It assesses different methods, such as Jacobian matching and contrastive representation learning, in terms of their effectiveness in transferring mechanisms and overcoming simplicity bias.

**Strenghts**

The paper’s exploration and assessment of various methods like Jacobian matching and contrastive representation learning is robust. It gives a comprehensive view of how different methods fare in the transfer of mechanisms.

The paper contains insights detailing how the effects of different loss terms vary depending on the dataset and modality and distribution shifts. This nuanced exploration adds depth to the paper's analysis.

The paper does a commendable job highlighting the issue of simplicity bias. It details how various methods impact this aspect, bringing a nuanced understanding of its implications in the knowledge distillation process.

**Weaknesses**

While the paper discusses trade-offs such as computational costs and KL divergence, a more structured and detailed analysis comparing these trade-offs across different methods would enhance the paper’s comprehensiveness.

The paper could benefit from a clearer presentation of the experimental setup to help readers better evaluate the robustness and reliability of the findings.

**Suggestions for Improvement**

Consider verifying the typos and misspellings.

An extended discussion on the role and impact of various loss terms, providing deeper insights and guidance on their effectiveness and applicability across different scenarios, would be a valuable addition.

**Conclusion**

The paper offers valuable insights into mechanism sharing in knowledge distillation, bringing a nuanced perspective on methodological effectiveness and simplicity bias. While it provides a robust discussion and evaluation, there's room for enhancement in aspects like trade-off analysis and clarity in experimental details to make the findings more comprehensive and actionable. The suggestions for future work pave the way for exciting directions in understanding model processing and vulnerability.

---

### Official Review · Reviewer_rFL2 · 2023-10-23
**Interesting problem setup, hard to follow presentation**

**Rating:** 5
**Confidence:** 4

**Review:**

The paper "What Does Knowledge Distillation Distill?" investigating what knowledge during the knowledge distillation process is transferred from teacher to student.

Pros:
- The question of what knowledge is being transferred during knowledge distillation is interesting and worth of further study.
- The paper consider three different formulation of knowledge distillation (KL of soft outputs, Jacobian matching, and contrastive representation distillation).
- The paper considers experiments from two different modalities (images and text).

Cons:
- My main issue with the paper is poor presentation of materials. It is relatively easy to follow the paper up to end of section 3. However, section 4 is very difficult to follow. The datasets, tasks, training and evaluation protocols are poorly explained. Results and conclusions are not easy to understand. Correspondence of presented results in section 4 to listed contributions in section 1 is not clear.
- Tables and figures are using several abbreviations that are hard to follow and not well explained.

---

### Official Review · Reviewer_axts · 2023-10-24
**Strong paper exploring knowledge transfer in knowledge distillation**

**Rating:** 8
**Confidence:** 4

**Review:**

This paper explores the question of how knowledge is transferred between a teacher and student in knowledge distillation. Rather than simply comparing the accuracy between the teacher and student, this work aims to characterize the similarity in mechanism, or whether the models rely on the same predictive attributes to make a decision. They compare three knowledge distillation techniques on synthetic datasets aimed to capture different mechanisms. They demonstrate that contrastive loss leads to the most similar mechanisms between teacher and student, and Jacobian loss additionally leads to more similar mechanisms than baseline (standard KL) distillation. Contrastive loss also avoids reliance on spurious simple mechanisms moreso than baseline and Jacobian loss.

This paper is quite strong and deserves acceptance into the workshop. They describe an evaluation technique and run comprehensive experiments. The results demonstrate the variance in distillation methods lending themselves to mechanism transfer and open the door for future work in the space. This is an important angle to consider in the space of knowledge distillation and interpretability.

---

### Official Review · Reviewer_XVaW · 2023-10-24
**Good analysis of knowledge distillation techniques**

**Rating:** 7
**Confidence:** 3

**Review:**

The authors provided a clear analysis of knowledge distillation methods trying to define the idea of knowledge transfer.
Ways of generating synthetic data, training and evaluation are explained in a good way.
Moreover, I appreciate the idea that the analysis was conducted on multiple data modalities (image and language).
The conclusion about which method performs better is expected, however the overall discussion is great.

---

### Decision · Program_Chairs · 2023-10-28

Accept (Poster)